

# Controls on Greenland moulin geometry and evolution from the Moulin Shape model

Lauren C. Andrews[1], Kristin Poinar[2,3], Celia Trunz[4]

[1]Global Modeling and Assimilation Office, NASA Goddard Space Flight Center, Greenbelt, MD, 20771, USA
[2]Department of Geology, University at Buffalo, Buffalo, NY, 14260, USA
[3]Research and Education in eNergy, Environment and Water (RENEW) Program, University at Buffalo, Buffalo, NY, 14260, USA
[4]Geosciences Department, University of Arkansas, Fayetteville, AR, 72701, USA

**Correspondence**: Lauren C. Andrews (lauren.c.andrews@nasa.gov)

**Abstract.** Nearly all meltwater from glaciers and ice sheets is routed englacially through moulins, which collectively comprise approximately 10–14% of the efficient englacial–subglacial hydrologic system. Therefore, the geometry and evolution of moulins has the potential to influence subglacial water pressure variations, ice motion, and the runoff hydrograph delivered to the ocean. We develop the ***Mou**lin **Sh**ape* (MouSh) model, a time-evolving model of moulin geometry. MouSh models ice deformation around a moulin using both viscous and elastic rheologies and models melting within the moulin through heat
dissipation from turbulent water flow, both above and below the water line. We force MouSh with idealized and realistic surface melt inputs. Our results show that variations in surface melt change the geometry of a moulin by approximately 30% daily and by over 100% seasonally. These size variations cause observable differences in moulin water storage capacity, moulin water levels, and subglacial channel size compared to a static, cylindrical moulin. Our results suggest that moulins are significant storage reservoirs for meltwater, with storage capacity and water levels varying over multiple timescales.
Representing moulin geometry within subglacial hydrologic models would therefore improve their accuracy, especially over seasonal periods or in regions where overburden pressures are high.

## 1 Introduction

Surface-sourced meltwater delivered to the glacier bed influences the evolution of the subglacial hydrologic system and associated subglacial pressures (e.g., Iken and Bindschadler, 1986; Müller and Iken, 1973). The efficiency of the subglacial
system, in turn, changes the flow patterns of the overlying ice on daily, seasonal, and multi-annual timescales (e.g., Hoffman et al., 2011; Iken and Bindschadler, 1986; Moon et al., 2014; Tedstone et al., 2015; Williams et al., 2020). Thus, glacial hydrology is a crucial factor in short-term calculations of mass loss on glaciers and ice sheets (Bell, 2008; Flowers, 2018). On the Greenland Ice Sheet, surface meltwater can take multiple paths, depending on its spatial origin. In the accumulation zone, meltwater may percolate through snow and firn, remaining liquid (Forster et al., 2014) or refreezing (MacFerrin et al., 2019).
In the ablation zone, meltwater runs over bare ice, coalesces into supraglacial streams, and pools into supraglacial lakes (e.g.,





Smith et al., 2015). These surficial water features – rivers, streams, lakes, aquifers, etc. – direct meltwater into englacial features that can deliver the water to the bed of the ice sheet (Andrews et al., 2014; Das et al., 2008; Miège et al., 2016; Poinar et al., 2017; Smith et al., 2015). Englacial features include moulins, which are near-vertical shafts with large surface catchments (~1–5 km$^2$ per moulin, Banwell et al., 2016; Colgan and Steffen, 2009; Yang and Smith, 2016), and crevasses, which are linear

features with limited local catchments (~0.05 km$^2$ per crevasse, Poinar et al., 2017). Together, moulins and crevasses constitute the englacial hydrologic system.

Water fluxes through the englacial system, and therefore to the subglacial system, are non-uniform in space and time. Variations in subglacial water fluxes can modify the form of the subglacial hydrologic system and influence ice motion on diurnal to multi-year timescales (Hoffman et al., 2011; Tedstone et al., 2015). Quantifying these temporal variations in water

fluxes to the glacier bed requires understanding the time evolution of the supraglacial and englacial water systems that deliver it. Ongoing research is making great strides in characterizing the supraglacial water network (Germain and Moorman, 2019; Smith et al., 2015; Yang et al., 2016). For instance, field observations from Greenland indicate that much of the supraglacial water network terminates into crevasses and moulins (Colgan et al., 2011a; McGrath et al., 2011; Poinar et al., 2015; Smith et al., 2015) and that these englacial features are important modulators of surface melt inputs to the ice sheet bed (Andrews et al.,

2014; Cowton et al., 2013).

Our knowledge of moulin sizes, scales, and time evolution has largely been informed by exploration and mapping of the top tens to hundred meters of a few moulins (Benn et al., 2017; Covington et al., 2020; Gulley et al., 2009; Holmlund, 1988; Moreau, 2009). These sparse field data indicate that moulin shapes deviate greatly from simple cylinders. Furthermore, deployments of tethered sensors into Greenland moulins have encountered irregularities, including apparent ledges, plunge

pools, and constrictions, below the depths of human exploration (Andrews et al., 2014; Covington et al., 2020; Cowton et al., 2013). These direct near-surface and indirect deep observations suggest that moulin geometry evolves a high degree of complexity at all depths.

State-of-the-art subglacial hydrology models are forced by meltwater inputs that enter the system through crevasses or moulins. These models generally represent the geometry of moulins in a simplified and time-independent manner, for

instance as a static vertical cylinder (e.g., Hewitt, 2013; Hoffman et al., 2016; Werder et al., 2013) or cone (Clarke, 1996; Flowers and Clarke, 2002; Werder et al., 2010). The basis for the cylindrical simplification arises from the assumption that depth-dependent variations in moulin size are small relative to the vertical scale of the moulin. The basis for time independence is the assumption that the moulin capacity is, again, small relative to that of the subglacial system. However, neither of these assumptions have been tested. Here, we explore the extent to which time evolution of moulin geometry affects the rate of

subglacial meltwater input and subglacial pressure in channelized regions of the bed.

We present the Moulin Shape (MouSh) model, a new, physically based numeric model that evolves moulin geometry over diurnal and seasonal periods (Fig. 1). The MouSh model can be coupled to subglacial hydrology models to more completely characterize the time evolution of the englacial and subglacial hydrologic systems, which are intimately linked.



## 2 Methods

### 2.1 The role of moulins in Greenland meltwater transport

Modeling of the subglacial hydrologic system has revealed the importance of moulins in the initiation of subglacial channels under both alpine glaciers (Hoffman and Price, 2014) and ice sheets (Werder et al., 2013). Thus, moulins comprise an important component of the overall glacial hydrologic system but are usually omitted or represented simply in subglacial hydrology models.

We estimate the relative path length of moulins within the overall glacial hydrologic system by combining available maps of moulin locations (Andrews, 2015; Hoffman et al., 2018; Smith et al., 2015) and maps of subglacial hydraulic potential (Fig. 2a & 3a-b), following Shreve (1972):

$$\Phi = k\rho_i g(s - b) + \rho_w gb \tag{1}$$

Here, $\Phi$ is the subglacial potential; $\rho_i$ and $\rho_w$ are ice and water density, respectively; $g$ is gravitational acceleration; and $s$ and $b$ are the surface and bed elevations, respectively, derived from available datasets (Howat et al., 2014; Morlighem et al., 2017, 2018). $k$ is a spatially uniform parameter that represents the fraction of flotation; that is, the ratio of water pressure and ice overburden pressure, where $k$=1 indicates that water pressure is equal to ice overburden pressure at the bed, and $k$=0 indicates that water is at atmospheric pressure (e.g., Rippin et al., 2003; Shreve, 1972). In reality, $k$ will not be uniform in space or time, nor indicative of whether there are subglacial channels; however, studies suggest that using $k$ in flow routing improves prediction in regions with large subglacial water fluxes that would likely organize into efficient or channelized flow (Chu et al., 2016; Everett et al., 2016; Fried et al., 2015; Palmer et al., 2011). We use $k$=1 to approximate the likely channelized subglacial pathways.

We calculate theoretical flow accumulation via the hydraulic potential gradient in two separate areas of the Greenland Ice Sheet: the Pâkitsoq and Russell Glacier regions (Fig. 3). Flow accumulation maps do not directly indicate whether flow is channelized or unchannelized, especially at far inland locations, where efficient subglacial drainage may still occur in the absence of channels (Meierbachtol et al., 2013). Thus, we define a flow accumulation area threshold of ~1 km$^2$, which is similar to but generally smaller than the size of individual supraglacial drainage basins. As subglacial channels are generally initiated at moulin locations (Werder et al., 2013), this ensures that we do not underestimate potential subglacial channel lengths. Furthermore, we only interpret subglacial accumulation pathways seaward of the 1,300-meter surface elevation contour as channelized. This threshold is supported by GPS analysis in the Pâkitsoq region (Andrews et al., 2018) and dye tracing in the Russell Glacier region (Chandler et al., 2013). Using this information and the distance between each moulin location and the nearest subglacial channel node, we estimate the total length of subglacial channelization in these two regions. We next extract the ice thickness at each moulin from the BedMachine dataset (Fig. 3c, Morlighem et al., 2018), following Andrews (2015) and Yang and Smith (2016). The total subglacial channel length, including the distance between moulins and the nearest flow pathway plus the sum of moulin depths, are used to estimate the percentage of the total subsurface (englacial–subglacial) path length that moulins comprise.



## 2.2 Physical moulin model

We develop the Moulin Shape (MouSh) model, a numeric model of moulin evolution that considers ice deformation and ice
melt associated with the dissipation of energy from turbulently flowing meltwater (Fig. 1). We include here a detailed
description of the model framework and each module that influences the time-evolving geometry of the modeled moulin.

### 2.2.1 Moulin geometry coordinate system

We discretize our model in the vertical ($z$) and radial ($r_1$ and $r_2$) directions, treating the moulin as a stack of egg-shaped (semi-
circular, semi-elliptical) holes in the ice that both change in size and move laterally relative to each other. We calculate moulin
geometry ($r_1$ and $r_2$) and water level ($h_w$) with a 5-minute timestep. Model calculations are performed in cylindrical coordinates,
where $\Pi(z)$ is the perimeter of the semi-circular, semi-elliptical moulin, using Ramanujan's approximation:

$$\Pi \approx \pi r_1 + \frac{1}{2}\pi[3(r_1 + r_2) - \sqrt{(3r_1 + r_2)(r_1 + 3r_2)}\,]  \qquad (2)$$

Here, $r_1$ and $r_2$ are the major and minor radii, respectively, for each node in the vertical direction.

We calculate the cross-sectional area of the semi-circular, semi-elliptical moulin as follows:

$$A_m = \frac{\pi r_1}{2}(r_1 + r_2) \qquad (3)$$

The plan-view orientation of the radii and the coordinate system, as detailed on a remotely sensed moulin, are indicated in
Fig. 2.

### 2.2.2 Ice deformation modules

We represent the deformation of the ice with the simplest possible combination of elastic and viscous components: a Maxwell
rheology, where elastic and viscous deformation occur independently, without interaction (Turcotte and Schubert, 2002). The
Maxwell timescale is equal to $(Y \times A \times \tau^2)^{-1}$, or roughly 10–100 hours for typical Greenland ice. We approximate this as ~1 day.
On timescales shorter than the Maxwell timescale, ice deformation is primarily elastic. On longer timescales, viscous
deformation dominates.

#### 2.2.2.1 Elastic deformation

Field measurements indicate that, nearly universally during the melt season, the water level in a moulin varies at a sub-hourly
timescale (Andrews et al., 2014; Covington et al., 2020; Cowton et al., 2013; Iken, 1972). This variability is shorter than, but
comparable to, the ~1-day Maxwell timescale for ice; therefore, we must assume that elastic deformation plays a role in the
response of the ice to variations in moulin water level.

The stress and deformational patterns around a borehole have been well studied in the rock mechanics literature
(Amadei, 1983; Goodman, 1989; Priest, 1993). We base our description of the stress field surrounding the moulin on that of a
fluid-filled borehole in a porous rock medium, described by Aadnoy (1987) and based on the Kirsch equations, which describe
stresses surrounding a circular hole in a rigid plate (Kirsch, 1898). We assume plane strain and approximate our moulin as a



stack of such plates with analogous holes (Goodman, 1989). A subtle difference is that our moulin shape is not circular, but egg-shaped: half circular, half elliptical.

130        At each vertical level $z$ in the moulin, we apply Hooke's Law to the stress field to calculate the strain, in horizontal cross-section, at all points on the moulin wall and in the surrounding ice. We then integrate these strains from an infinite distance (cylindrical coordinate $r = \infty$) to the moulin wall ($r = r_m$). The result is the expected elastic displacement, $\Delta r_E$, of a segment of a moulin with radius $r_m$:

$$\Delta r_E \; = \; \frac{r_m}{Y}\left[(1 + \nu)(P - \tfrac{1}{2}(\sigma_x + \sigma_y) \; + \; \tfrac{1}{4}(\sigma_x - \sigma_y)(1 - 3\nu - 4\nu^2) + \tfrac{1}{4}\tau_{xy}(2 - 3\nu - 8\nu^2)\right] \tag{4}$$

Here, $Y$ is Young's modulus for uniaxial deformation, $\nu$ is Poisson's ratio, and $\sigma_x$, $\sigma_y$, and $\tau_{xy}$ are the background deviatoric and shear stresses that describe the regional setting of the moulin (typically compressive and of order 100 kPa). We apply this equation to both moulin radii, the semi-circular radius $r_1$ and the semi-elliptical radius $r_2$, separately. The outward pressure in the moulin $P$ is the difference between the hydrostatic and cryostatic pressures:

$$P \; = \; \rho_w g(h_w - z) - \rho_i g(H_i - z) \tag{5}$$

where $h_w$ is the height of the water above the bed, $H_i$ is the ice thickness, and $z$ is the vertical coordinate. When water is above the flotation level, the moulin elastically opens at all depths. When the water level is below flotation, which is the typical case, elastic deformation closes the moulin at all depths.

       Because we use a flat bed for all model runs presented, we define $h_w$ as 'moulin water level', which only includes the hydrostatic water head, as opposed to 'moulin hydraulic head', which would include both the hydrostatic and elevation heads.

**2.2.2.2 Viscous deformation**

Over times longer than the ~1-day Maxwell timescale, viscous deformation is the dominant deformation process. We represent total viscous deformation in two modes: (1) radial opening and closure of the moulin, which changes the size of the moulin (Section 2.2.2.2.1), and (2) vertical shear of the moulin, which changes the shape but not the size of the moulin (Section 2.2.2.2.2).

**2.2.2.2.1 Radial opening and closure**

Moulins close when they lose their water source at the end of a melt season (Catania and Neumann, 2010). Similarly, boreholes close if they are not filled with drilling fluid with a density similar to ice (Alley, 1992). Our modeled moulin is intermediate to these edge cases, since it typically contains water. When the moulin is filled with water to the flotation level, it will stay open at its base and viscously close at and below the water level. When water is above flotation, the moulin will viscously

open at all depths. When the water level is below flotation, which is the typical case, viscous deformation shrinks the moulin at all depths.

       We sum the glacio-static stress, $\sigma_i$, and the hydrostatic stress, $\sigma_w$, to get the total depth-dependent stress, $\sigma_z$, at all levels $z$ within the moulin:



$$\sigma_i = -\rho_i g(H_i - z) \tag{6a}$$

$$\sigma_w = \rho_w g(h_w - z) \tag{6b}$$

$$\sigma_z = \sigma_i + \sigma_w \tag{6c}$$

In this formulation, positive stress causes outward expansion of the moulin walls (radial growth), and negative stress reduces the size of the moulin (radial closure).

We calculate strain rate ($\dot{\varepsilon}$) from the total depth-dependent stress (Eq. 6c) using Glen's Flow Law:

$$\dot{\varepsilon} = E \, A(T,P) \cdot \left(\tfrac{1}{3}\sigma_z\right)^3 \tag{7}$$

where $E$ is an enhancement factor, and $A(T,P)$ is the flow law parameter. For the flow law parameter, we use the standard relationship from Cuffey and Paterson (2010, Equation 3.35), which is a function of temperature $T$ and pressure $P$.

We follow borehole studies by Naruse et al (1988) and Paterson (1977) to write strain, $\varepsilon$, in the radial direction as

$$\varepsilon = ln\left(\tfrac{r_f}{r_0}\right) \tag{8}$$

where a moulin with initial radius $r_0$ and final radius $r_f$ underwent radial strain of $\varepsilon$.

We use the time derivative of Eq. (8) to calculate the moulin radius in the $(i+1)^{th}$ timestep as a function of the radius in the $i^{th}$ timestep, separated by time $\Delta t$:

$$r_{i+1} = r_i \, exp(\dot{\varepsilon} \, \Delta t) \tag{9}$$

with strain rate given by Eq. (7). This is the same relationship used by Catania and Neumann (2010).

### 2.2.2.2.2 Shear deformation

We use Glen's Flow Law to calculate the change in shape of the moulin due to regional-scale ice flow. This deforms the entire moulin in bulk, shearing it in the vertical and shifting it laterally downstream, without changing its radii.

We calculate $u_d(z)$, the rate of deformational ice flow in the downstream direction, from ice temperature $T$ and pressure $P$, surface slope $\alpha$, a constant enhancement factor $E$, and ice thickness $H_i$, using Glen's Flow Law (Cuffey and Paterson, 2010):

$$u_d = \left|2 \, E \, (\rho_i \, g \, \alpha)^n \int_{z=b}^{z=s} A(T)(H - z)^n dz\right| \tag{10}$$

We obtain ice deformation rates of ~20 m yr[-1], which is typical of the ablation zone in western Greenland (Ryser et al., 2014).

### 2.2.3 Phase change modules

The second mode that changes the geometry of the moulin is ice ablation from or accretion to the moulin walls. During the melt season, the flow of water into and through the moulin generates turbulence, which as it dissipates acts to melt back the moulin walls, expanding the size of the moulin. There is also a small component of melting due to temperature differences between the water and surrounding ice. Outside the melt season, conduction of latent heat into the surrounding ice causes stagnant water to freeze back onto the moulin walls, contracting the size of the moulin.



### 2.2.3.1 Refreezing

We calculate freeze-on when water flow within the moulin is laminar (Reynold's number $Re < 1$), which occurs only outside
the melt season.

The MouSh model conserves energy via a balance of sensible heat within the ice sheet and the latent heat of the water
in the moulin. The model does not track sensible energy fluxes associated with water flow through the moulin. Equivalently,
we assume that all water in the moulin is at the melting point, and that all sensible heat changes within the ice in the moulin
walls are compensated by latent heat transfer via the refreezing of moulin water onto the moulin wall. We calculate the
wintertime freeze-on thickness, $\delta$, at each timestep and at each depth $z$ below the moulin water level as follows:

$$\delta(z) = \frac{k_i}{\rho_i L_f} \frac{\partial T}{\partial x} \Delta t \tag{11}$$

Here, $k$ is the thermal conductivity of ice, $L_f$ is the latent heat of freezing, and $\frac{\partial T}{\partial x}$ is the lateral temperature gradient in the ice
at the moulin walls. This temperature gradient varies in space and time and entirely dictates the refreezing rate. We calculate
the temperature of the ice sheet, $T(x,z)$, at each depth $z$ using the one-dimensional, diffusion-only heat equation with Dirichlet
boundary conditions for far-field temperature, which we define at 30 meters from the moulin, and 0°C at the moulin wall. We
treat the latent heat deposition at the moulin wall as a source term (Poinar et al., 2016). We treat each depth independently,
thus assuming zero vertical diffusion or advection. We calculate the change in moulin water volume from freezing, $V_{frz}$, by
summing the refrozen ice thickness, $\delta$, around the perimeter of the moulin at all depths $z$, and converting ice volume to water
volume:

$$V_{frz} = \frac{\rho_i}{\rho_w} \int_0^{h_w} \Pi(z)\delta(z)\, dz \tag{12}$$

We also include an option to model refreezing rates more economically, following Alley (2005):

$$\delta(z) = 2\frac{\Delta T}{L_f} \sqrt{\frac{k\, C_p}{\pi \rho_i}} \left(\sqrt{t} - \sqrt{t - \Delta t}\right) \tag{13}$$

where $\Delta T$ is the depth-varying difference between the far-field temperature and 0°C, and $C_p$ is the specific heat capacity of
ice.

### 2.2.3.2 Moulin wall melting

During the melt season, turbulent energy dissipation from water flowing through the moulin melts back the moulin walls. We
calculate the thickness of ice melted, $m(z)$, and add this new volume of meltwater to the water already in moulin, similarly to
Eq. (12) for $V_{frz}$:

$$V_{wallmelt} = \frac{\rho_i}{\rho_w} \int_0^{h_w} \Pi(z)m(z)\, dz \tag{14}$$

The dissipation of turbulent energy and the associated melting of the surrounding ice will increase the local moulin radius. We
parameterize turbulence in two separate spatial domains: (1) within the water column of the moulin (Section 2.2.3.2.1) and (2)
above the water level along the side of the moulin, as supraglacial input falls to the water level (Section 2.2.3.2.2). The second





parameterization, while simplifying complex hydraulics and melting patterns, is necessary to offset the steady viscoelastic closure of the moulin above the water line.

The parameterizations of turbulently driven melting we use in both regimes rely on two simplifications. First, the volume of water moving through each vertical model node is constant within each time step. This ensures that water mass is conserved and that all model elements below the water line are water filled; however, this eliminates the potential long-term storage of meltwater within plunge pools caused by non-uniform incision into the ice. Second, all energy generated from turbulent dissipation is instantaneously applied to melting the surrounding ice. This neglects any heat transport within the

water, which is a common approximation in subglacial models (e.g., Hewitt, 2013; Schoof, 2010; Werder et al., 2013), and allows us to make the simplifying assumption that meltwater entering the moulin is at 0°C and at the pressure melting temperature at all points below the water line. As part of our melt parameterization, we also include the effect of the temperature difference between the water and the surrounding ice (Jarosch and Gudmundsson, 2012) because, unlike for subglacial channels, we cannot assume that the surrounding ice is at the pressure melting temperature. Though we model instantaneous

heat exchange, we note recent findings that the appropriate heat transfer coefficient can be difficult to determine (Sommers and Rajaram, 2020).

### 2.2.3.2.1 Melt below the water line

Below the waterline, the vertical velocity of the water is dictated by the hydraulic gradient within the system and the local cross-sectional area of the moulin. Under such conditions, head loss – the departure of the hydraulic head from that calculated

by Bernoulli's equation – reflects the energy dissipated as heat. We parameterize head loss using the Darcy–Weisbach equation, which relates water velocity ($u_w$) to changes in the hydraulic gradient ($dh_l/dl$, head loss per unit length along flow), via the hydraulic diameter ($D_h$), a dimensionless friction factor ($f_R$) and gravitational acceleration ($g$). Because water velocity is constrained by mass balance within the system, we calculate the head loss ($dh_l/dl$), or turbulent energy dissipated into melting the moulin walls, as follows:

$$\frac{dh_l}{dL} = \frac{u_w^2 f_R}{2 D_h g} \tag{15}$$

The differential element $dL$ represents the path length over which the water experiences head loss:$dL = \sqrt{dx^2 + dz^2}$ for horizontal distance $dx$ and vertical drop $dz$. The friction factor ($f_R$) is a unitless model parameter that controls the rate of head loss within the system. Its value thus directly affects the amount of internal melting. Most subglacial models fix the Darcy–Weisbach friction factor, with values ranging from 0.01 to 0.5 (e.g., Colgan et al., 2011b; Schoof, 2010; Spring and Hutter,

1981) or use equivalent values of Manning's $n$ (e.g., Hewitt, 2013; Hoffman and Price, 2014). Such constant values, however, are somewhat at odds with field observations that indicate highly variable subglacial channel roughness over a range of time and spatial scales (Gulley et al., 2014; Mankoff et al., 2017). Alternatively, other models parameterize channel roughness using a geometry-dependent friction factor (e.g., Boulton et al., 2007; Clarke, 2003; Flowers, 2008). All current parameterizations for time-varying roughness, however, were developed outside moulins and thus may not accurately represent conditions there,





where observations of extensive ice scalloping, for instance, make moulins distinct from other conduits (Gulley et al., 2014).
      Thus, MouSh has options for fixed or variable friction factors. The overall effect on moulin geometry is modest.

         For time and geometry dependent parameterization of roughness, we choose the Bathurst parameterization (Bathurst,
      1985):

$$f_R = \left( -1.987 \ log \left( \frac{k_s}{5.15 R_h} \right) \right)^{-2}$$
(16)

Here, $k_s$ is the surface roughness height and $R_h$ is the channel hydraulic radius, which is equal to the cross-sectional area divided
      by the wetted perimeter at heights at or below the water line. Because we approximate the moulin as a half-circular, half-
      elliptical cylinder, the hydraulic radius $R_h$ of a water filled node is

$$R_h = \frac{A_m}{\Pi}$$
(17)

      for moulin perimeter $\Pi$. The Bathhurst parameterization produces a range of friction factors over approximately two orders of
magnitude for typical Greenland moulins. This and other roughness parameterizations cannot adequately account for changing
      form roughness, including sinuosity and large-scale ice scalloping. To explore this, we complete a sensitivity study (Sect. 2.3
      & 3.2) where we fix the friction factor over the expected range, centered on $f_R = 0.1$. We use a constant $f_R = 0.1$ for all other
      model runs presented.

         We calculate the head loss used to determine the amount of moulin wall melting using a simple energy balance
equation:

$$\rho_i C_w \Delta T \frac{dS}{dt} + \rho_i L_f \frac{dS}{dt} = \rho_w g Q \frac{dh_L}{dL}$$
(18)

      The first term represents sensible heat, the small amount of energy needed to warm the surrounding ice to the pressure melting
      point. Otherwise, Eq. (18) follows previous work based in temperate ice conditions (e.g., Gulley et al., 2014; Jarosch and
      Gudmundsson, 2012; Nossokoff, 2013). Equation (18) can be rearranged and modified for our elliptical system such that the
change in moulin radius due to melting is:

$$\Delta r_t = \left[ \frac{\rho_w g Q}{\Pi \rho_i (C_w \Delta T + L_f)} \frac{dh_L}{dz} \right] dt$$
(19)

      Here, $dh_L/dL$ is the head loss over the model node (Eq. 15), $\Pi$ is the wetted perimeter of a water-filled moulin node; $Q$ is the
      discharge from the moulin-subglacial system as dictated by the subglacial model component (Section 2.2.4.2); and $\Delta T$ is the
      temperature difference between the water (prescribed to be at the pressure melting point) and the surrounding ice, which can
be assigned a number of different profiles from Greenland as described in Table 1.

**2.2.3.2.2 Melt above the water line**

Above the water line, a range of complex processes drive melting. A first-principles approach is to quantify melting due to the
potential energy loss of falling water, following the work on terrestrial waterfalls (e.g., Scheingross and Lamb, 2017).





However, nearly all waterfall-based parameterizations rely on abrasion between waterborne sediment and the substrate as the
primary mechanism of erosion. Instead, we implement a simple parameterization for open-channel flow with the understanding
that the complexities of thermal erosion are not completely captured. In our model, open-channel melting occurs only on the
up-glacier wall of the moulin and follows two ad-hoc rules based on the slope between the vertical nodes: (1) open-channel
turbulent melting is applied if the slope of the upstream moulin wall allows water to flow over it; and (2) a small, prescribed
amount of melting is applied when the upstream wall slope is vertical or overhung, because while water cannot flow directly
along the ice, spray and other processes likely drive some amount of melting. These cases are respectively (1) the open-channel
zone and (2) the falling water zone.

*Open-channel zone:* In the open-channel zone, we use a similar approach as for melting below the water line.
However, the hydraulic radius is adjusted to reflect the observation that water runs down only one wall of the moulin, and
higher friction factors is used to represent complex geometries (e.g., Covington et al., 2020). Due to the presence of a
discontinuity between open-channel and water-filled regions (at the water line), we parameterize the wetted perimeter as
follows:

$$R_{h_{open}} = \pi r_1 \tag{20}$$

To drive melting in the open-channel region, we use Eq. (17) with a hydraulic radius of only a portion of the moulin. Note that
the hydraulic radius prescribed for open-channel flow is likely larger than the small region over which water is flowing in the
natural system (Fig. 2d), minimizing the amount of melt in the model.

*Falling water zone*: In the falling water zone, there is very limited interaction between the moulin walls and the water.
For simplicity, we assume that a small fraction, $f_w$, of the potential energy lost as water falls reaches the moulin walls, perhaps
impacting it as spray, and is used to melt the surrounding moulin. The change in radius due to this process is as follows:

$$\Delta r_f = f_p \frac{(\rho_w/\rho_i)gQ_{in}}{L_f \Pi} dt \tag{21}$$

We set $f_p$, the fraction of potential energy applied to melting the surrounding walls, to 1% for the model runs presented here.

**2.2.4 Water flux into and out of the moulin**

We enforce conservation of water mass within the moulin system as follows:

$$\Delta V_{water} = (Q_{in} - Q_{out})\Delta t + V_{wallmelt} - V_{frz} \tag{22}$$

The change in the volume of water stored in the moulin, $\Delta V_{water}$, within a time step is proportional to the volumetric discharge
into and out of the moulin ($Q_{in}$ and $Q_{out}$, respectively) and the volume of water added due to melting ($V_{wallmelt}$) or removed
due to refreezing ($V_{frz}$). The latter two terms are described above in Eq. (14) and (12), respectively. Discharge into and out of
the system are described below.




**2.2.4.1 Meltwater runoff from the ice-sheet surface**

We force the MouSh model with time-varying water inputs from the supraglacial environment, $Q_{in}$. We use two different $Q_{in}$
scenarios: a simple diurnal cosine with maximum and minimum discharges ranging between ~1 and 5 m³s⁻¹, in rough
agreement with observations near the margins (Eq. 23, Chandler et al., 2013; McGrath et al., 2011; Smith et al., 2017); and
realistic supraglacial discharge over a melt season, determined by using in-situ surface melting data and internally drained
catchment size and geometry (Yang and Smith, 2016).

We use the following cosine curve to represent our simplest form of supraglacial discharge into the moulin during
sensitivity studies:

$$Q_{in} = cos(\pi(t - 19.5)/12) + 3 \tag{23}$$

Here, $t$ is time in days and $Q_{in}$ is in m³s⁻¹. This function has its daily peak at 19:30 hours and a daily minimum at 07:30.

Initial surface runoff values for the 2019 melt season were modified using a synthetic unit hydrograph derived for the
ablation zone and parameters appropriate for western Greenland (Table 2, Smith et al., 2017). The parameters for the unit
hydrograph were determined during the middle of the melt season and therefore may inaccurately represent routing delays at
the beginning and end of the melt season.

The MouSh model can also accept base flow directly to the subglacial module. We design base flow as a loose
approximation of additional subglacial water flow from varied upstream sources, including other moulins on the same
subglacial channel, regional basal melt, and the addition and removal of meltwater from subglacial storage. The latter reflects
the englacial void ratio used in many subglacial models. Base flow is generally required to maintain realistic moulin water
levels. In the moulin runs forced by realistic $Q_{in}$, we represent subglacial flow from ~5 surrounding moulins by prescribing
base flow as five times the running 5-day mean of $Q_{in}$. This application mimics the seasonal evolution of surface melt and
maintains a slightly larger subglacial channel than would otherwise occur, which reduces otherwise unrealistically large daily
swings in modeled moulin water level.

**2.2.4.2 Water flow to the subglacial system**

We couple the moulin model and a single evolving subglacial channel controlled by melt opening and creep closure (Covington
et al., 2020; Schoof, 2010) using a reservoir-constriction model (Covington et al., 2012) that simulate flows between the two
elements.

The time rate of change of water level ($h$) is developed in Covington et al. (2012):
$$\frac{dh}{dt} = \frac{1}{A_m(h)}(Q_{in} + Q_{base} - Q_{out}) \tag{24}$$

where $A_m(h)$ is the moulin cross-sectional area at the water level (Eq. 3), $Q_{in}$ is the influx of surface meltwater, $Q_{base}$ is the
base flow at each time step (Sect. 2.2.4.1), and $Q_{out}$ is the meltwater output from the subglacial channel, defined as follows:

$$Q_{out} = c_3 S^{5/4} \Psi / \sqrt{|\Psi|} \tag{25}$$



Here, $S$ is the subglacial channel cross-sectional area. The hydraulic gradient $\Psi = -\rho_w g \frac{d(h_w+b)}{dL}$ assumes zero bed slope and

a linear gradient in the moulin water level ($h_w$) to the outlet at a horizontal distance $L$, where the pressure head is zero. Finally, $c_3$ is a flux parameter:

$$c_3 = \frac{2^{5/4}}{\pi^{1/4}} \sqrt{\frac{\pi}{(\pi+2)\rho_w f_r}}. \tag{26}$$

Equation (26) for $c_3$ follows Covington et al. (2020), who corrected a small error from the original Schoof (2010) formulation.

We use an equation from Schoof (2010) for the time rate of change in subglacial channel cross-section area $S$, with

the first part describing the turbulent melting of the subglacial channel walls, and the second term describing closure due to the pressure of the overlying ice:

$$\frac{dS}{dt} = c_1 Q_{out} \Psi - c_2 N^n S \tag{27}$$

Here, the constant $c_1 = \frac{1}{\rho_i L_f}$ with $\rho_i$ the ice density and $L_f$ the latent heat fusion of ice, the constant $c_2 = 2An^{-n}$ with the Glen's

flow law parameters $A \approx 10^{-24} \text{Pa}^{-3}\text{s}^{-1}$ and $n = 3$. The effective pressure $N = P_i - \rho_w g h_w$.

Replacing $Q_{out}, \Psi$, and $N$ in Eq. (27) yields

$$\frac{dS}{dt} = c_1 c_3 S^{5/4} \left(\frac{\rho_w g h}{L}\right)^{3/2} - c_2 (p_i - \rho_w g h)^n S \tag{28}$$

Equations (24) and (28) are numerically solved simultaneously. The parameters used in this module are included in Table 1 and are the same as those used in the englacial component of MouSh, with the exception of the flow law parameter $A$. In the englacial system, $A$ is calculated from local temperature within the ice column, which can be as cold as -23°C in our study area

(Iken et al., 1993). This contrasts with the temperature at the ice-bed interface, which must be at the melting point; thus, the subglacial component of MouSh uses higher $A$ values.

### 2.2.4.3 Water volume contributions from melting and freezing

Water balance within the moulin and the subglacial channel is dictated by recharge from a supraglacial stream ($Q_{in}$), discharge through a subglacial channel ($Q_{out}$), and any change in volume due to melting or refreezing (related to $m(z)$, the radial melt or

refreezing rate, in m s⁻¹), such that the volume of water in the moulin ($V_m$) is

$$\frac{dV_m}{dt} = Q_{in} - Q_{out} + \int_0^H \Pi(z)\, m(z)\, dz \tag{29}$$

The integral term varies in space and time, with high melt rates above the water line during the melt season (when $Q_{in} > 0$), and moderate melt rates at and below the water line during and after the melt season, when there is water flow through the moulin ($Q_{out} > 0$) and refreezing below the water line throughout the winter (when $Q_{in} = Q_{out} = 0$).

### 2.3 Sensitivity to uncertain parameters

We explored the sensitivity of our results to the values of seven parameters, shown in Fig. 4, with the prescribed ranges shown in Table 1. We studied the effect on water level; the moulin radius at the equilibrium water level; the volume and water storage





of the moulin; and cross-sectional area of the subglacial channel at the end of a ten-day model run. These values reach equilibrium, with daily oscillations superimposed, after 3–5 days. We also tested the dependence of our results on the initial moulin radius, $R_0$, which we varied across an order of magnitude from 0.5 to 5.0 meters.

We varied the value of a uniform deformation enhancement factor $E$ over an order of magnitude ($E = 1$ to 9), which affects viscous flow of the ice surrounding the moulin. We also tested the effect of ice temperature, independent of the enhancement factor. We used five different temperature profiles: cold ice temperatures (mean ~ -15°C, range -23.1°C to the pressure melting point) measured in the center of Jakobshavn Isbræ (Iken et al., 1991); moderate ice temperatures (mean ~ -7 °C, range -13.5°C to the pressure melting point) measured at the GULL site in Pâkitsoq (Lüthi et al., 2015; Ryser et al., 2014); warmer ice temperatures (mean ~ -5°C, range -9.3°C to the pressure melting point) measured at the FOXX site in Pâkitsoq (Lüthi et al., 2015; Ryser et al., 2014); a hypothetical linear profile from -5°C at the surface to 0°C at the bed; and, finally, a fully temperate ice column. These different ice temperature scenarios affected the creep closure rates of ice through the temperature-dependent softness parameter $A$ by approximately a factor of 6 from the coldest profile (Iken et al., 1993) compared to the fully temperate column.

We also examined moulin sensitivity to elastic deformation by varying Young's modulus ($Y$) of the ice column between 1–9 GPa (e.g., Vaughan, 1995). We also tested sensitivity to the values of friction factors for the moulin walls. These factors control melt rates associated with the dissipation of turbulent energy. MouSh has two friction factors: $f_M$ (moulin friction; below the water line) and $f_{oc}$ (open-channel friction; above the water line). We varied these friction factors across two orders of magnitude. We did not vary the subglacial friction factor. Finally, we varied values for basal ice softness over two orders of magnitude and independently examined moulins over a range of ice thicknesses (670–1570 m) and corresponding distance from the terminus (~20–110 km).

## 2.4 Sensitivity to local conditions

We examined moulins over a range of ice thicknesses, corresponding distance from the terminus, and appropriate $Q_{in}$ forcings for three different representative locations on the ice sheet (Table 2). We designed this suite of model experiments to provide general guidance on the range of variability in moulin geometries over the course of the 2019 melt season and over a range of supraglacial catchment sizes. As part of this analysis, we examine season-long and daily differences in model outputs and variation in each model component (viscous, elastic and phase change) and their relative importance in driving moulin geometry change.

## 2.5 Comparison to a cylindrical moulin

Subglacial models generally use a time-invariant vertical cylinder to represent moulins. To investigate and quantify the efficacy of our time-evolving moulin shape model, we drove MouSh and a static cylinder with the same meltwater inputs. We use the time-mean radius at the water level as the radius of the static cylinder; this is 1.4 m for Basin 1 and 1.3 m for Basin 2. We compared the resulting moulin water level, moulin capacity, subglacial cross-sectional area and meltwater input difference

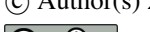



400 (due to melt generated within the model itself) across these runs. We compared the moulin water level values directly (*variable water level – cylindrical water level*) and compared other metrics by percentage difference (2 (*variable – cylindrical*) / (*variable + cylindrical*)).

## 3 Results

### 3.1 Moulins as part of the channelized englacial–subglacial system

405 The capacity for water storage in the englacial system, relative to the subglacial system, depends on both the spatial density of moulins and the volumes of individual moulins. Below 1300 m elevation, moulin densities are ~0.27 per km$^2$ in Pâkitsoq (270 moulins) and ~0.16 per km$^2$ (704 moulins) in the Russell region (Fig. 2a, 3a–b). The total length of subglacial flow pathways are 765 km and 4,679 km for Pâkitsoq and Russell regions, respectively. The distribution of ice thicknesses at each moulin location are shown in Fig. 3c; the cumulative englacial path lengths are ~110 km in Pâkitsoq and ~446 km in the Russell 410 region. Thus, we find that moulins comprise between 10% (Russell) and 14% (Pâkitsoq) of the path length that water takes from its entry to the englacial-subglacial system to its exit at the terminus or calving front. These lengths are not insignificant and suggest that moulin geometry and evolution may be important to subglacial processes.

### 3.2 Sensitivity of MouSh to parameter values and deformational processes

A range of ice characteristics affect the time evolution of moulin geometry. These include the initial moulin size, temperature 415 and viscosity of the ice column, viscosity of basal ice, friction factors, and ice thickness. These factors are either highly spatially variable (e.g., ice thickness) or poorly known (e.g., basal ice viscosity). We quantify the effect of these factors on moulin water level and moulin volume, moulin geometry, and subglacial channel cross-sectional area over both multi-day and diurnal timescales by performing multiple independent sensitivity studies (Section 2.3).

We find that moulins reach sizes near equilibrium within 3–5 days, regardless of their initial radii, but that the 420 equilibrium aspect ratio is sensitive to the initial radius (Fig. 4a–b). Moulins initialized with large radii (5 m) reach equilibrium shapes that are more elliptical than moulins initialized with more realistic radii (0.5 m). In particular, the equilibrium major radius is 36% larger and the minor radius is 27% larger for 5 m initial conditions compared to 0.5 m (Fig. 4a), which manifests as a 14% increase in moulin capacity (Fig. 4b). The initial condition also affects the magnitude of diurnal variations in major radius (97%), minor radius (26%), and moulin water storage (13%), with larger-initialized moulins varying less (Fig. 5a). 425 Regardless of initial condition, the major radius undergoes lower-magnitude diurnal fluctuations than the minor radius. The initial moulin radius does not significantly affect equilibrium subglacial channel size (0.04%), moulin water level (0.02%), or moulin water storage (1.5%). Diurnal variations in subglacial channel size (0.3%), water level (0.4%), and moulin volume (5.4%) were all also insensitive to initial conditions (Fig. 5b).

Three major parameters affect the degree of viscous and elastic deformation in the moulin: the ice flow enhancement 430 factor $E$, the ice temperature profile $T(z)$, and Young's modulus $Y$. We tested a span of reasonable values representative of





Greenland ice (Table 1) and found a limited effect on moulin geometry. Equilibrium moulin water level, subglacial channel area, and their diurnal variabilities remain constant (<0.1% change) over the tested range of these parameters (Fig. 4d,f,h & 5d,f,h). Moulin capacity and water storage show moderate sensitivity (~20% in equilibrium value and ~40% in diurnal range) across the range of $E$ and $T$ scenarios tested; a decrease in moulin capacity and water storage pair with an increase in the diurnal

variability for these variables. For instance, varying $E$ across an order of magnitude grew the equilibrium major radius by 23% and shrank the equilibrium minor radius by 44%, with a net effect that moulins had 23% less volume and 20% less water storage capacity in softer ice ($E = 9$) compared to harder ice ($E = 1$) (Fig. 4c–d). Similarly, the different ice temperature profiles we tested caused variations of 11% in moulin major radius, 18% in moulin minor radius, and 24% in moulin capacity and moulin water storage, with warmer ice hosting smaller moulins (Fig. 4e–f).

We varied Young's modulus, $Y$, across one order of magnitude. With the highest Young's modulus, moulin major radius increased by 50% compared to the lowest, minor radius decreased by 15%, moulin water volume increased by 38%, and moulin capacity increased by 56% (Fig. 4g–h). The equilibrium water level decreased insignificantly (<0.1%) and the subglacial channel area increased insignificantly (<0.1%) across this range of $Y$. These effects are comparable to those of $E$, which we also varied over one order of magnitude, and $T$, which changed the englacial flow-law parameter $A$ by approximately

a factor of 6.

    In contrast to the above parameters, we find that moulin geometry is strongly sensitive to the choice of basal ice softness (prescribed $A_{basal}$) and the friction factors used within the moulin ($f_m$ and $f_{oc}$). Melting due to the dissipation of turbulent energy is partially controlled by the friction factors chosen for the moulin walls. The friction factor above the water line ($f_{oc}$, "open channel") does not significantly affect moulin water level (<0.1% change for $f_{oc}$ variations over two orders of

magnitude), moulin volume (4%), moulin water storage (2%), or subglacial channel area (<0.1%) over either long or diurnal timescales (Fig. 4m–n and 5m–n). However, like the deformational parameters, the open channel friction factor does affect moulin radii, with the major radius growing by 36% as the open channel friction factor increases over two orders of magnitude, and the minor radius decreasing by 27%. This dampens the diurnal variability in both radii.

    Increasing the friction factor below the water line ($f_m$) had similar effects to changing $f_{oc}$. Increasing $f_m$ by two orders

of magnitude increased the cross-sectional area of the moulin by 106%, via a 15% increase in the major radius and a 95% increase in the minor radius. The water volume increased by 116% and the storage capacity increased by 100% (Fig. 4k–l) while the equilibrium water level and the subglacial channel area changed by <0.1%. Increasing the underwater friction factor $f_m$ also increased the diurnal variability of the moulin capacity and water storage (Fig. 5k–l) by increasing the diurnal differential melt rate.

The two parameters which have the largest impact on moulin water level are the basal ice softness ($A_{basal}$) and the moulin location on the ice sheet, described jointly by the ice thickness ($H$) and distance from the terminus ($L$). We varied basal ice softness by two orders of magnitude. Softer basal ice increased the size and storage capacity of the moulin: the major radius by 21%, the minor radius by 25%, the total capacity by 88%, and the stored water volume by 112% (Fig. 4i–j). These changes also increased the equilibrium water level by 57% and the subglacial channel area by 24%, unlike tests on englacial parameters



($E$, $T$, and $Y$), which did not affect the water level and subglacial channel area. These changes occur because softer basal ice increases the rate of subglacial creep closure, which reduces subglacial channel cross-sectional area, which reduces water throughflow in the moulin and increases water level, which in turn reduces the amount of viscous and elastic radial closure in the moulin. Increasing the basal ice softness $A_{basal}$ to approximately $10^{-23}$ Pa$^{-3}$s$^{-1}$ increases the diurnal variability in the sizes of the subglacial channel and moulin (Fig. 5i–j); however, increasing $A$ above this value causes moulin water levels to rise high

enough that diurnal fluctuations are truncated by the ice thickness. Results at these values are therefore not shown on Fig. 4. The precise value of $A$ where this transition occurs depends on other traits of the moulin, including ice thickness.

       We co-varied ice thickness and distance from terminus using a parabolic approximation for a perfectly plastic ice surface profile (Cuffey and Paterson, 2010). Variations in ice thickness from 670 to 1570 m (80%) increase equilibrium subglacial conditions by 20% and increase equilibrium water levels by 107% (Fig. 4o–p). Increasing ice thickness and distance

from the terminus increases the moulin major and minor radii by 4%, increases moulin volume by 97%, and increases moulin water storage by 114% (Fig. 4p). We also find significant increases in diurnal variability in subglacial channel size (28%), water level (105%), moulin radii (major radius 85% and minor radius 22%), moulin volume (130%), and moulin water storage (140%) in thicker ice farther from the terminus (Fig. 5o–p).

       Overall, we find that MouSh-modeled moulins are primarily sensitive to the friction factors for water flow through

the moulin, basal ice softness, and location on the ice sheet (ice thickness and distance from the terminus). The results are less sensitive to englacial material factors that govern elastic and viscous deformation. The observed sensitivity to the ice thickness and distance from terminus signals that moulin geometry can vary spatially. The sensitivity to friction factors and basal ice softness indicates that the values of these poorly constrained parameters should be carefully chosen and kept in mind when interpreting model output.

**3.2.2 Contributions to moulin shape**

Figure 6 illustrates the role of each process that changes moulin radius under equilibrium conditions: phase change, viscous deformation, and elastic deformation. We use standard values for all parameters (Table 1) and we vary ice thickness and distance from terminus. We find that moulin shape is quite similar across different ice thicknesses, while mean water level, moulin capacity (Fig. 6a–e) and the diurnal range in moulin radius (Fig. 6g) increase with ice thickness. We also analyze

temporal variations in each process (Fig. 6g). The times of maximum melt and maximum viscous closure are slightly offset, with peak melting occurring during the most rapid decline in viscous deformation (Fig. 6g). This offset aligns with the rising limb of the input hydrograph, when the moulin is small and increases in $Q_{in}$ raise water level and, in turn, elevate englacial melt rates and reduce viscous deformation.

       Melt rates both above and below the water line contribute to moulin growth (Fig. 6f–g). Melt above the water level

occurs due to stream or waterfall erosive processes, which in MouSh occur only within a fraction of the total circumference (Fig. 2d), which manifests as growth of the major radius. The actual rate of melting, however, is also dictated by the area over





which water flow occurs, which under our parameterization is related to the cross-sectional area of the moulin at any given depth (Fig. 6f).

Elastic deformation, like viscous deformation, closes the moulin except when the water level is above flotation. Elastic deformation rates are generally smaller than viscous rates, except between ~100–300 meters above the bed, where viscous deformation is minimized by cold ice temperatures (Lüthi et al., 2015; Ryser et al., 2014). Diurnally, elastic deformation varies with a similar pattern to viscous deformation, though over less range.

**3.3 Moulin shape in different environments**

We modeled the seasonal growth and collapse of moulins in a range of environments across the Greenland Ice Sheet using
realistic melt forcings derived for the 2019 melt season (Section 2.4). These model runs varied with respect to ice thickness, moulin distance from the terminus, base flow, and the magnitude, diurnal range, and seasonal evolution of supraglacial inputs (Table 2; Fig. 7a). Overall, we find that moulin setting affects the scale of diurnal and seasonal variability in the size and water capacity of moulins as well as the evolution of subglacial channels (Fig. 7 and 8).

The sizes of all three modeled moulins reach equilibrium with the melt forcing within 10 days of the onset of the melt
seasons (Fig. 7b–c). As the water flux increases over the next few weeks, each moulin grows in response to increasing supraglacial inputs, both diurnally and with a long-term trend (Fig. 7c), though this growth is more significant in thicker ice (Fig. 8m–p). The subglacial channel grows with a similar pattern, but interestingly, the setting and fluxes of Basin 1 and Basin 2 result in very similar subglacial channel cross-sectional areas (Fig. 7d). However, moulin water level shows diurnal variations superimposed on a relatively constant base value throughout the melt season (Fig. 7b).

Though the three moulins all evolve in a similar fashion, there are differences in moulin water capacity, equilibrium water level (Fig. 7), overall moulin geometry (Fig. 8), and the magnitude of englacial deformation (Fig. 9). Diurnal variation in moulin capacity is slightly larger in thicker ice, due to higher rates of deformation and greater melt rates within both the moulin (Fig. 9a). While the relative fraction of phase change (melting) to deformation is generally near 1, in the thickest ice, the absolute value of total deformation is generally higher than that of the phase change term. This indicates that viscous and
elastic opening play a role in maintaining moulin geometry when water levels are high (Fig. 9b), otherwise, the moulin would continually close.

The ratio of elastic to viscous deformation generally ranges from ~0.4 to ~0.8, depending on ice thickness (Fig. 9b). Elastic deformation rates in the moulin depend on a linear function of ice thickness, while viscous rates are related to ice thickness cubed.  Thus, at lower elevations, the elastic contribution is maximized (~0.8 of viscous deformation), while at high
elevations, significant increases in viscous closure lowers the relative contribution of elastic deformation (~0.4 of viscous deformation). This increase in viscous closure in thick ice also minimizes subglacial channel size in thick inland ice (Fig. 7d), despite water levels persistently at or near the ice surface.





Each moulin has a different equilibrium capacity (Fig. 7c). This, in addition to differences in supraglacial inputs, ensures that daily moulin water level variations are substantially different across moulins. Basin 1 exhibits the largest variation in moulin water level, followed by Basin 2 (Fig. 10a). Basin 3 shows the lowest daily change; however, this is due at least in part to the fact that water overtops the moulin nearly daily (Fig. 7b and 8m–n). Changing water levels drive changes in moulin and subglacial capacity. Over the melt season, daily change in moulin capacity can be as low as 5% during lulls in diurnal melt
variability (Basin 3) or as high as 32% following a recovery from a low melt day (Basin 1; Fig. 10b). However, in general all moulins display a similar daily change in capacity of ~10–25%, with no clear pattern as to which elevation experiences greater short-term changes in moulin capacity.

        The subglacial system undergoes diurnal variations in channel size between 1 and 21% (Fig. 10c). These changes are similar in magnitude to daily capacity changes within the moulin but exhibit more variability across ice thicknesses and are
related to the daily changes in moulin water level (Fig. 10a). This suggests that the time evolution of moulin geometry dampens the diurnal pressure fluctuations that drive subglacial channel growth and collapse. Evidence for this can be seen in the temporal pattern of moulin water level and subglacial channel cross-sectional area (Fig. 10a and 10c). To test this idea, we compared results from static and time-evolving moulins (Section 3.4).

### 3.4 Comparison to cylindrical moulins

To examine the role moulin evolution plays in modifying the subglacial hydrologic system, we compared moulin water levels, moulin capacity, and subglacial channel size between model runs with a fully evolving moulin and runs with a static cylindrical moulin. We performed these tests with realistic melt inputs based on the 2019 melt season (Section 2.4), at moulins with low and moderate ice thicknesses (553 m – Basin 1 and 741 m – Basin 2). We defined the radius of the fixed cylinder as 1.4 m and 1.3 m for Basin 1 and 2, respectively. This results in fixed cross-sectional areas (~6 m$^2$ and ~5 m$^2$) within the range of the
spatially invariant moulin cross-sectional areas ~2–10 m$^2$ often prescribed in subglacial models (e.g., Andrews et al., 2014; Banwell et al., 2013; Bartholomew et al., 2012; Cowton et al., 2016; Meierbachtol et al., 2013; Werder et al., 2013). Inter-comparison of these runs allows us to examine the role moulin geometry has on subglacial pressures (Covington et al., 2020; Trunz et al., in review).

        Comparison of water level, moulin capacity, moulin water storage, and subglacial cross-sectional area between fixed
and evolving moulins show differences on both the diurnal and seasonal times scales (Fig. 11). Moulin water levels (*variable – fixed*) can be substantial (Fig. 11a), with short term differences driven by variable melt conditions reaching a maximum of -129 m (Basin 1) and -177 m (Basin 2). The long-term daily average differences are -28 m and -49 m for Basin 1 and Basin 2, respectively. These differences are driven by a combination of differences in moulin capacity and subglacial channel size (Fig. 11b-c) and are despite a total increase in the meltwater input relative to a fixed moulin, due to melt generated from turbulent
dissipation (Fig. 11d).

        Generally, the evolving moulin is larger, stores more water, generates more water through internal melting, and maintains a larger subglacial channel (Fig.11b–d), which all contribute to the observed difference in water levels. Midway





through the melt season, the evolving moulin exhibits capacities and storage volumes 10–50% larger than those of the fixed

cylinder (Fig. 12b). As meltwater inputs taper at the end of the melt season (day ~97 in Fig. 11b), the capacity and storage

volume in the evolving moulin falls below that of the fixed cylinder, whose volume does not adjust in response to the forcings.

This seasonal evolution is consistent between the two ice thicknesses tested.

The capacity differences between the variable and fixed moulin contribute directly to dampening the supraglacial

input signal and dampening of moulin water levels. This contributes directly to an increase in subglacial channel size (Fig.

11d), both diurnally and over the season. The seasonal difference between the variable and fixed moulin forcing is relatively

constant, though punctuated by dips associated with reduced moulin water level differences (Fig. 11a).

## 4 Discussion

### 4.1 Formation mechanisms of moulins

Moulins can form through multiple processes, including cut and closure of supraglacial streams (Gulley et al., 2009) and

vertical hydrofracture through cold ice (Das et al., 2008). The formation mechanism dictates the initial geometry, which then

evolves rapidly in response to a range of ice and melt processes to reach equilibrium geometry.

On the Greenland Ice Sheet, moulin locations are generally disassociated with crevasse fields (Colgan et al., 2011a;

Phillips et al., 2011; Smith et al., 2015) and, instead, occupy regions of compressional stresses, including supraglacial lake

basins (Catania et al., 2008; Poinar and Andrews, 2020; Stevens et al., 2015). Episodic local water inputs to the bed can

instigate transient stresses that exceed the fracture toughness of ice, forming crevasses in otherwise-compressional regimes

(Christoffersen et al., 2018; Hoffman et al., 2018; Poinar and Andrews, 2020). Depending on the meltwater influx, crevasses

that form moulins may propagate to the bed in less than a day (Stevens et al., 2015) or over multiple years (Poinar, 2015).

Regardless of formation timescale, all known moulins on the Greenland Ice Sheet are thought to be vertical or near vertical.

We consider the formation timescales of moulins in the context of the shape evolution of a mature moulin. Using

MouSh, we find that in the absence of external forcing, such as time-variable $Q_{in}$, the size of a moulin reaches its equilibrium

value in ~1–10 days. This relaxation time is comparable to the Maxwell time for ice (10–100 hours), as expected for a linear

visco-elastic system. Our relaxation time also compares well to the equilibration timescale defined by Covington et al. (2020)

for their modeled moulin – subglacial conduit system, which Trunz et al. (in review) found to be 1–20 days. The most

realistically sized moulins in Trunz et al. (in review) had relaxation times closer to 1 day. Their modeled channel system was

governed solely by melt and viscous deformation and lacked elastic deformation; this may explain their modestly longer

relaxation time compared to ours.

If the process of moulin formation occurs on a timescale shorter than the 1–5-day relaxation time, the formation

process likely will not influence the overall form of the englacial system at equilibrium. This time range includes hydrofracture

during rapid lake drainage (~2 hours) and slow lake drainage (<~6 days, e.g., Selmes et al., 2011), and likely also the

reactivation of existing moulins in ensuing melt seasons, which, based on the timing difference between surface melt onset




and ice acceleration, occurs on a multi-day timescale (Andrews et al., 2018; Hoffman et al., 2011; Zwally et al., 2002). On the other hand, moulin formation by cut-and-closure occurs over years to decades (Gulley et al., 2009), well above the MouSh relaxation time and the Maxwell time for ice. The interdependence of formation and evolution of these moulins gives us less confidence in applying our model to moulins with cut-and-closure origins. Those moulins primarily occur in temperate near-surface ice within polythermal glaciers (Gulley et al., 2009) and have not been reported on the Greenland Ice Sheet.

**4.2 Comparison of modeled and observed moulin geometries**

Field observations suggest that moulin geometry evolves a high degree of complexity. Observations include anecdotes of difficulty deploying sensors to the bottom of a moulin, which suggests the presence of kinks, ledges, knickpoints, and other twists (Andrews et al., 2014; Covington et al., 2020; Cowton et al., 2013). Complex geometry revealed during mapping moulins above the water line further suggests that moulins are not simply vertical cylindrical shafts (Covington et al., 2020; Moreau, 605 2009).

The MouSh model suggests that the energy transfer from turbulent meltwater entering the moulin to the surrounding ice drives highly spatially variable melt rates above the water line. We incorporated the open-channel melt module to allow a large opening to emerge above the water line (Fig. 6a–e and 8). When we run MouSh without the open-channel module (Sect. 2.2.4.3), the surface expression of the moulin is much smaller than observed and in some cases, the moulin will pinch closed 610 at the ice-sheet surface.

The value of the open-channel friction factor and the size of the spatial footprint over which melting occurs directly affects the size of the upper, air-filled chamber of the moulin. MouSh consistently predicts ledges at the top and bottom of a consistent diurnal range in water level. Thus, we infer that energetic subaerial water flow drives formation of moulin complexity above the water line, and diurnal fluctuations around a steady multi-day water level drive ledge formation through 615 underwater melting and visco-elastic deformation. Energetic water flow is commonly observed at stream-fed moulins near the peak of the melt season (Pitcher and Smith, 2019) or during and immediately following rapid lake drainage (Chudley et al., 2019). This suggests that complex moulin geometries form during periods of relatively consistent water supply. Conversely, multi-day rises in water level, driven by either the surface water supply ($Q_{in}$) or the basal water supply (baseflow), can erase geometric complexities such as ledges, as seen in MouSh results during a melt event (Fig. 8).

Above the water line, explored moulins in Greenland show highly variable shapes from moulin to moulin (e.g., Covington et al., 2020). Some moulins, for example the FOXX moulin, are nearly cylindrical within the explored depth (~100 m), with radii comparable to what we model (~2 meters). Others, like the Phobos moulin, open some tens of meters below the surface to large caverns with radii approaching 10 meters, a similar morphology to karst caves with narrow entrance shafts. MouSh can produce large openings above the water line if we use a suitably large open channel friction parameter, although 625 we lack a narrow entrance shaft. These differences are due to the inability of model parameterizations to represent complex geometries such as scalloping, plunge pools and knickpoint migration (Gulley et al., 2014; Mankoff et al., 2017). Indeed, instead of modeling processes above the water line as turbulent open flow, they could be modeled using geomorphic





parameterizations to model waterfall migration, perhaps resulting in the clearer development of steps and plunge pools. However, these parameterizations are generally based on contact abrasion and debris cover (e.g., Scheingross and Lamb, 2017), making the translation to thermal erosion difficult.


Below the water line, MouSh results indicate that a cylinder is a reasonable representation for newly formed moulins in Greenland. However, there are two caveats. First, moulin cross-sectional area, and thus water storage capacity, can vary substantially over the course of a day or season (Fig. 11). Second, in instances where moulins are reactivated over multiple melt seasons (Chu, 2014; Smith et al., 2017), there can be substantial deformation, as suggested by cable breakage in boreholes (Ryser et al., 2014; Wright et al., 2016).


Observations show a wide range of moulin volumes above the water line, and moulin volumes predicted by MouSh are sensitive to the value of the open-channel friction factor. Given the flexibility of model results, we should continue to rely on field exploration to measure moulin size and geometry above the water line and make efforts to constrain the parameters that affect sub-seasonal growth and collapse. MouSh results below the water line are less sensitive, so we rate underwater exploration of Greenland moulins at a lower priority. Overall, results from the MouSh model demonstrate that moulin geometry evolves substantially over diurnal to seasonal timescales and varies with ice conditions.


## 4.3 Diurnal water level oscillations and moulin size

Moulin geometry can directly alter the relationship between meltwater inputs and moulin water level changes – the primary driver of subglacial channel evolution (Andrews et al., 2014; Cowton et al., 2013). Field measurements of moulin water levels indicate diurnal oscillations of 3–12% (Covington et al., 2020), ~25% (Andrews et al., 2014) and >20% (Cowton et al., 2013) of overburden pressure. These diurnal fluctuations are larger than those observed in boreholes, which are generally thought to sample inefficient components of the subglacial hydrologic system (Andrews et al., 2014; Meierbachtol et al., 2013; Wright et al., 2016).


Our model results agree well with observations of moulin water level: diurnal fluctuations of 15–25% of overburden pressure, with larger oscillations in thicker ice. To explain larger-than-expected daily oscillations (~10%) in thinner ice, Covington et al. (2020) incorporated moulin cross-sectional area as a free parameter into their model. Matching field measurements of water level required a modeled moulin radius of ~5 m (~75 m$^2$ cross-sectional area) at ice thickness 500 m and a much larger moulin (radius ~20 m and cross-sectional area ~1500 m$^2$) at ice thickness 700 m (Covington et al., 2020). For comparison, MouSh predicts average radii of ~1.3 to 1.4 m (~5 m$^2$ cross-sectional area) at these ice thicknesses using parameters described in Table 2, including substantially larger meltwater inputs compared to Covington et al. (2020). The drastic differences in moulin size despite similar variations in diurnal water level variability between our study and Covington et al. (2020) cannot easily be attributed to a single factor, but may be explained by our limited ability to model processes above the water line, our inclusion of baseflow, substantial differences in meltwater input, fluctuations in moulin capacity, or that their measured water levels were not from the same moulin they mapped englacially. Nevertheless, we observe substantial differences in water level between fixed and variable geometry moulins and at both elevations (Fig. 11). Water levels are less






variable and generally lower in the evolving moulins compared to the fixed cylindrical moulin. We also note that for both ice thicknesses, the fixed moulin frequently overtops while the evolving moulin does not (Fig 11a) for the same melt, ice, and subglacial conditions. Thus, to match observed moulin water level fluctuations without evolving the moulin geometry, a fixed cross-sectional area substantially larger than the associated subglacial channel may be necessary, as reported in Covington et al. (2020).

## 4.4 Moulin geometry and the englacial void ratio

Subglacial hydrology models use an englacial void ratio parameter to represent bulk storage and release of meltwater in the englacial system (see Flowers and Clarke (2002) for the best description). The englacial void ratio allows subglacial models to resolve observed diurnal fluctuations in water pressure and, if coupled to a dynamical ice model, corresponding diurnal variations in ice flow. This parameter represents bulk behavior and is usually set constant over the model domain, yet it must be tuned by comparing to local observations (e.g., Bartholomaus et al., 2011; Hoffman et al., 2016; Werder et al., 2013).

Recent work suggests that fluctuations in water level are controlled by the size of the moulin near the water level (Trunz et al., in review): moulins with larger cross-sectional areas have lower diurnal variability in water level, if given the same melt input. Furthermore, our results suggest that the amount of water stored in a moulin is highly dependent on local conditions, such as water pressure on daily to seasonal timescales, and ice thickness (Fig. 7). Thus, we explore the possibility that detailed model-based information on moulin sizes and shapes could inform the englacial void ratio used in subglacial hydrology models. This would allow time dependence and finer spatial variation, including in the vertical dimension as well as horizontal, than is currently possible with a bulk parameter. Periods of increased supraglacial inputs can require a sizable increase in englacial void ratio for subglacial models to accurately predict moulin water level (Hoffman et al., 2016). During these times, MouSh predicts rapid growth in moulin capacity (Fig. 7 and 8). This correspondence suggests plausible close ties between moulin size and the englacial void ratio: moulin size modifies englacial storage spatially and temporally.

MouSh can be used to infer both moulin size and shape, which would effectively change the englacial void ratio in all three spatial dimensions and time. The shape of the moulin imposes new temporal variability on water level and subglacial channel size: moulins with large near-surface chambers that funnel down to become narrower at the water line, for instance, have lower-magnitude and smoother variations in water level compared to cylindrical moulins, whereas moulins with small surface openings that widen toward the water line have larger and peakier water-level variations (Trunz et al., in review). Thus, when the shape of a moulin is explicitly resolved, any assumed linear relationship between melt input rates and the range or pattern of oscillations in water level and subglacial channel size breaks down. The relationship also changes with the water level in the moulin; hence it varies in time.

MouSh demonstrates that moulin capacity can vary greatly both seasonally and during short periods of large variability in supraglacial input. Moulin growth rates are largest particularly when water levels are above flotation, maximizing turbulent melting and outward visco-elastic deformation. Our results show that moulin capacity changes by ~20% daily (Fig. 10) and ~50–100% over the melt season (Fig. 7c), with larger changes during periods of large supraglacial input variability





and at locations with thicker ice. These variations in moulin shape and size may explain difficulties with modeling subglacial
behavior during melt events (Cowton et al., 2016), which are sometimes addressed by temporarily increasing englacial storage
(Hoffman et al., 2016). Our results with MouSh lead us to recommend that moulin shape and size be modeled alongside the
evolution of the subglacial system, especially during periods of large meltwater variability, in order to more accurately predict
subglacial water pressures and ice motion.

Practical limits on model complexity or computational costs may preclude fully time-evolving moulin geometries.
While not ideal, a static shape is still preferable to a static cylinder (Trunz et al., in review). Therefore, we interpret our moulin
shape results (Fig. 8) to recommend a representative shape for a static moulin. Below the water line, a cylinder is a reasonable
approximation, especially in thinner ice or for newly made moulins, for which full-column ice deformation is minimized.
Above the water line, moulin shape is widely variable in time, by location, and across parameter combinations. It is especially
sensitive to the friction parameter for open-channel flow (Fig. 4), with low friction values making bottle-shaped moulins that
have narrow necks above the water line and larger chambers below the water line, and high friction values making goblet-
shaped moulins with open rooms and amphitheaters above the water line atop a narrower geometry below the water line.
Exploration of Greenland moulins to date has uncovered multiple goblet-shaped moulins and a few instances of near-
cylindrical moulins, but no bottle-shaped moulins (Covington et al., 2020; Moreau, 2009; Trunz et al., in review). Overall, our
MouSh results support goblet-shaped moulins, although with great variation in the height and width of the upper chamber.

**4.5 The role of elastic deformation in moulin geometry**

Our model results indicate that the equilibrium moulin geometry is dictated by a balance of visco-elastic deformation and
turbulence-driven melting (Fig. 6 and 9). In both the sensitivity study and realistic model runs, visco-elastic deformation
generally closes the moulin, while melting of the surrounding ice consistently opens the moulin. The exception is when moulin
water levels exceed flotation, in which case all three mechanisms open the moulin. In all model runs, we find that elastic and
viscous deformation are of the same order of magnitude, and that the elastic mode can be between 40% and 80% of the viscous
deformation (Fig. 6g and 9). The importance of elastic deformation holds even in the bottom few hundred meters of the ice
column, where stress conditions are similar to those in subglacial models (Fig. 6f). However, the relative importance of viscous
and elastic deformation in closing the moulin is also dependent on the values of Young's modulus and viscous enhancement
factor (Fig. 6 & 9). Despite extensive study of these parameters, their values are difficult to constrain. Currently, the space of
viscous and elastic parameter values could conceivably allow either elastic or viscous deformation to dominate the closure of
a moulin. This underscores the importance of including both modes in the MouSh model.

Current subglacial hydrology models represent subglacial channel development (opening) by turbulent energy
dissipation and destruction (closing) by viscous deformation alone. Some more recent work involving elastically responding
storage elements or elastic flexure of the ice sheet has occurred (Clarke, 1996; Dow et al., 2015), and there have been efforts
to use elastic deformation or fluid compressibility to improve numeric stability of channel equations (Clarke, 2003; Spring and
Hutter, 1981, 1982). Interestingly, Clarke (2003) chose to use fluid compressibility due to model integration times. Yet, elastic





deformation has generally been omitted from current models of subglacial channelization, even when modeling rapid changes in meltwater inputs (< 1 day; e.g., Hewitt, 2013; Hoffman et al., 2016; Werder et al., 2013). This choice is likely because the role of elastic deformation was considered negligible over timescales of subglacial evolution (e.g., days to weeks). However, the importance of elastic deformation in diurnally closing moulins, particularly in thinner ice (Fig. 9b), suggests that its exclusion from subglacial channel models could result in the underestimation of channel closure rates when water levels are below flotation.

This leads us to ask why elastic deformation is absent from subglacial models, particularly because its importance relative to viscous deformation is difficult to constrain given the current range of observed Young's modulus (e.g., Vaughan, 1995). Hypothetical subglacial channel models that included elastic deformation alongside viscous deformation would show less temporal asymmetry, particularly in thinner ice, where channel closure may be strongly dictated by elastic deformation. Elastic-incorporating models would also likely predict larger diurnal variations in channel size and moulin water level. This in turn would incite stronger local pressure gradients at the bed, increasing connectivity between the channel and the surrounding distributed system.

## 4.6 Potential coupled englacial–subglacial hydrology models

Moulins occupy a moderate fraction of the channelized englacial–subglacial system and moulins persistently form along subglacial flow pathways (Fig. 3). The subglacial component of MouSh includes an optional baseflow term, which is necessary to produce realistic equilibrium water levels with the realistic supraglacial inputs we prescribed (Fig. 7). The baseflow value we used does not accurately represent any process because our model runs resolve only a single moulin connected to a single channel, whereas in the real world, multiple moulins feed a network of channels. Though idealized, the baseflow term conceptually connects to a number of potential water sources. These include (1) basal melting from geothermal and frictional heating, (2) supraglacial water delivered via nearby moulins that are connected to the same subglacial channel, and (3) water that moves from the channelized system to the surrounding inefficient system at high pressures and then flows back into the subglacial channel at lower water pressures (Hoffman et al., 2016; Mair et al., 2002, 2002; Tedstone et al., 2015).

Baseflow maintains a larger, less variable subglacial channel. This can alternately be achieved by lessening the local hydraulic gradient, thus increasing the mean water pressure along a given reach. This may locally occur where one subglacial channel enters another in an arborescent network (Fountain and Walder, 1998). MouSh currently does not have an interconnected network of channels; however, this is under development (Trunz et al., 2020).

Water inputs from surrounding moulins is likely the primary origin of base flow at low and moderate elevations. A substantial amount of baseflow, however, is needed to maintain high pressures necessary to keep a moulin open at high elevations (Fig. 7a): up to 30 $m^3 s^{-1}$. There are two potential explanations for this. First, supraglacial drainage basins at higher elevations are generally larger due to muted transmission of basal topography through thick ice (Crozier et al., 2018; Gudmundsson, 2003; Karlstrom and Yang, 2016). Thus, neighboring moulins deliver high fluxes to the subglacial channel. However, the wider spacing of inland moulins reduces their density within the network. The more likely explanation is that a





channelized system is a poor representation of subglacial drainage under thick ice, where lower surface slopes reduce the growth rate of subglacial channels, forcing most water flow through the inefficient system (Andrews et al., 2018; Chandler et al., 2013; Dow et al., 2014; Meierbachtol et al., 2013). Coupling MouSh to a more sophisticated subglacial hydrologic model that includes both the efficient and inefficient systems may therefore be necessary to accurately model moulins in thick ice.

We use a highly simplified model of the subglacial hydrology system: a single channel that connects the moulin to
the ice-sheet margin. Yet, MouSh results clearly indicate that including and evolving englacial component can reduce diurnal and long-term subglacial pressures by allowing moulins to behave as a time-varying storage mechanism, which has implications for subglacial channel growth and size (Fig. 11). Nevertheless, this model lacks a distributed system, which limits its fidelity for assimilating daily meltwater volumes into the subglacial system. Realistically, the channelized subglacial system cannot always accommodate the full volume of meltwater produced during summer days, and a portion of this water goes into
the distributed system (e.g., Mair et al., 2001, 2002). In our model, however, when the channelized system is overwhelmed, the water level in the moulin rises above what is typically observed, and sometimes even exceeds the height of the ice. The melt-driven opening and creep closure processes in the subglacial model explain this behavior: A lower water input to the moulin ($Q_{in}$) lowers the water flux into the subglacial system ($Q_{out}$), which lowers the melt rates that keep subglacial channels open, reducing the size of the subglacial channels and thus further reducing the subglacial water flux. This increases the water
level in the moulin. Thus, a reduced rate of surface melt can counterintuitively raise the modeled water level, whereas in reality, much of that water would enter the inefficient subglacial hydrologic system when moulin water levels exceed flotation. If the moulin model were coupled to a two-component subglacial model that represents the inefficient system alongside the channelized system, we would anticipate a much-improved ability to assimilate a wide range of meltwater input rates.

**5 Conclusions**

First results from the MouSh model show that moulins are not static cylinders. Their shapes oscillate daily by some 30% around an equilibrium value reached within the first week of the melt season. These daily fluctuations change the water volume held in the englacial hydrologic system, which in turn influences the evolution of the subglacial channels that moulins feed. When we represent a moulins a static cylinder in our englacial–subglacial hydrology model, these daily fluctuations go absent, and the overall volume of water stored in the englacial system is underestimated by 50–100%. Modeled moulin size and shape
may provide a more realistic representation for the englacial void ratio commonly used in subglacial hydrology models, particularly with future efforts to improve the parameterization of moulin development above the water line. This could be achieved by using an englacial hydrology – channelized subglacial system model, such as the MouSh model we present here, to characterize variability in moulin size and shape, or by coupling moulin models to more complete models of the subglacial system (channelized, distributed, and optionally weakly connected) to make a unified englacial–subglacial hydrology model
system. Improving the representation of the englacial–subglacial system to explicitly include moulins would have greatest efficacy during periods of rapidly varying supraglacial input (e.g., during beginning and end of the melt season and during



melt events) and in inland areas, with thick ice and high overburden pressures. These are coincident with situations where subglacial models without moulins, or with implicitly static moulins, tend to perform poorly.

*Data availability*. The Moulin Shape model is publicly available at https://github.com/kpoinar/moulin-physical-model/releases/tag/v1.0-MouSh-beta. The model results used in the analysis presented here are available within the above GitHub repository and are archived at the University at Buffalo Libraries at http://hdl.handle.net/10477/82587.

*Author contributions*. L.C.A. and K.P. jointly conceived of and developed the MouSh model. Both L.C.A. and K.P. designed
the study, executed the model runs, analyzed the data, produced the figures, and wrote the manuscript. C.T. implemented the subglacial module, participated in discussions, and edited the manuscript.

*Acknowledgements*. This work was supported by NASA Cryosphere grant 80NSSC19K0054 (L.C.A. and K.P.), the Global Modeling and Assimilation Office at NASA Goddard Space Flight Center funded under the NASA Modeling, Analysis, and
Prediction (MAP) program (L.C.A.), the Research and Education in eNergy, Environment and Water (RENEW) Institute at the University at Buffalo (K.P.), and the United States National Science Foundation award number NSF-ANS 1603835 (C.T.). We acknowledge DigitalGlobe, Inc. for providing WorldView images via the Enhanced View Web Hosting Services and the support therein provided by the Polar Geospatial Center under NSF-OPP awards 1043681 and 1559691.

*Competing interests*. The authors declare that they have no conflicts of interest.

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





1085

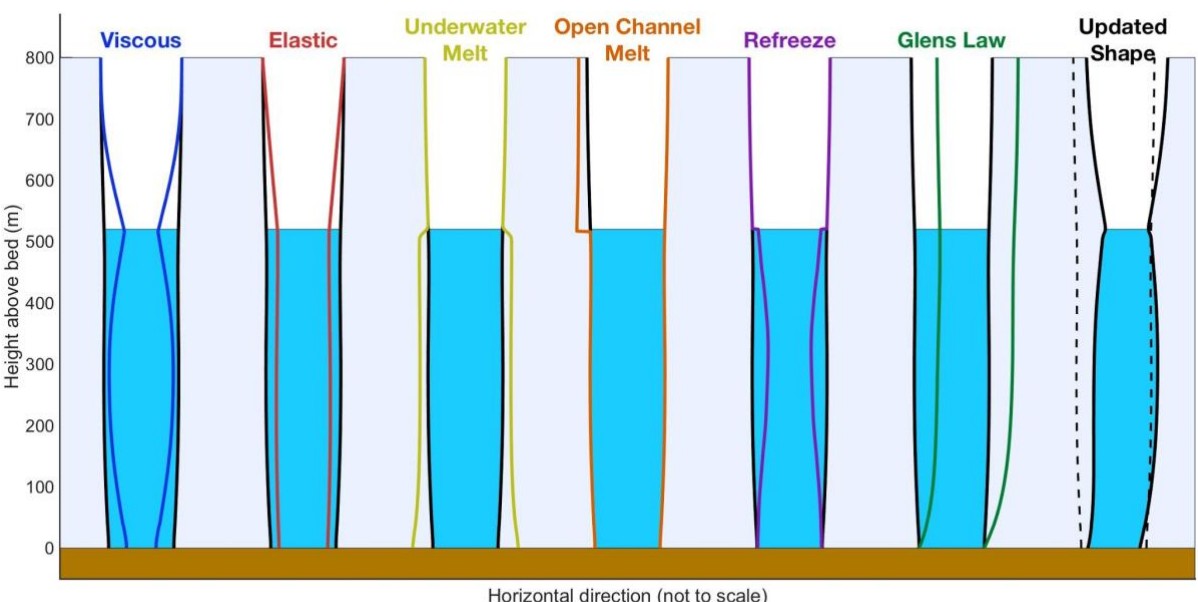

**Figure 1. Processes included in the MouSh model.** Black lines show a base moulin geometry that each process acts on, and colored lines show the change in moulin geometry (not to scale) due to that process alone. From left to right: changes in moulin geometry due to viscous deformation; elastic deformation; melting by turbulent energy dissipation of flowing water inside the moulin; melting by open-channel water flow along bare ice; refreezing over winter inside the moulin; and deformation due to ice motion prescribed by Glen's flow law. The right-most moulin shows the moulin geometry before (dashed black lines) and after (solid black lines and blue water) a hypothetical model timestep, i.e., the sum of all processes shown in the preceding panels. Changes are not to scale.



**Figure 2. Surface expression of a moulin and its reflection in the MouSh model.** (a) Mosaic of Sentinel-2 scenes from August 2019 (MacGregor et al., 2020) across the western Greenland Ice Sheet. Pâkitsoq region (green), Russell Glacier area (purple), and location of example moulin (yellow) are shown. (b) WorldView-2 scene from July 2010 of an approximately 1.2 × 0.8 km region surrounding the example moulin (yellow) formed by a drained supraglacial lake. (c) Detail of panel b, with the inflow stream and moulin indicated. (d) Detail of panel c, showing the moulin minor radius $r_1$, major radius $r_2$, and water input $Q_{in}$ from the inflow stream, as represented in the MouSh model. WorldView image © 2010 DigitalGlobe, Inc.


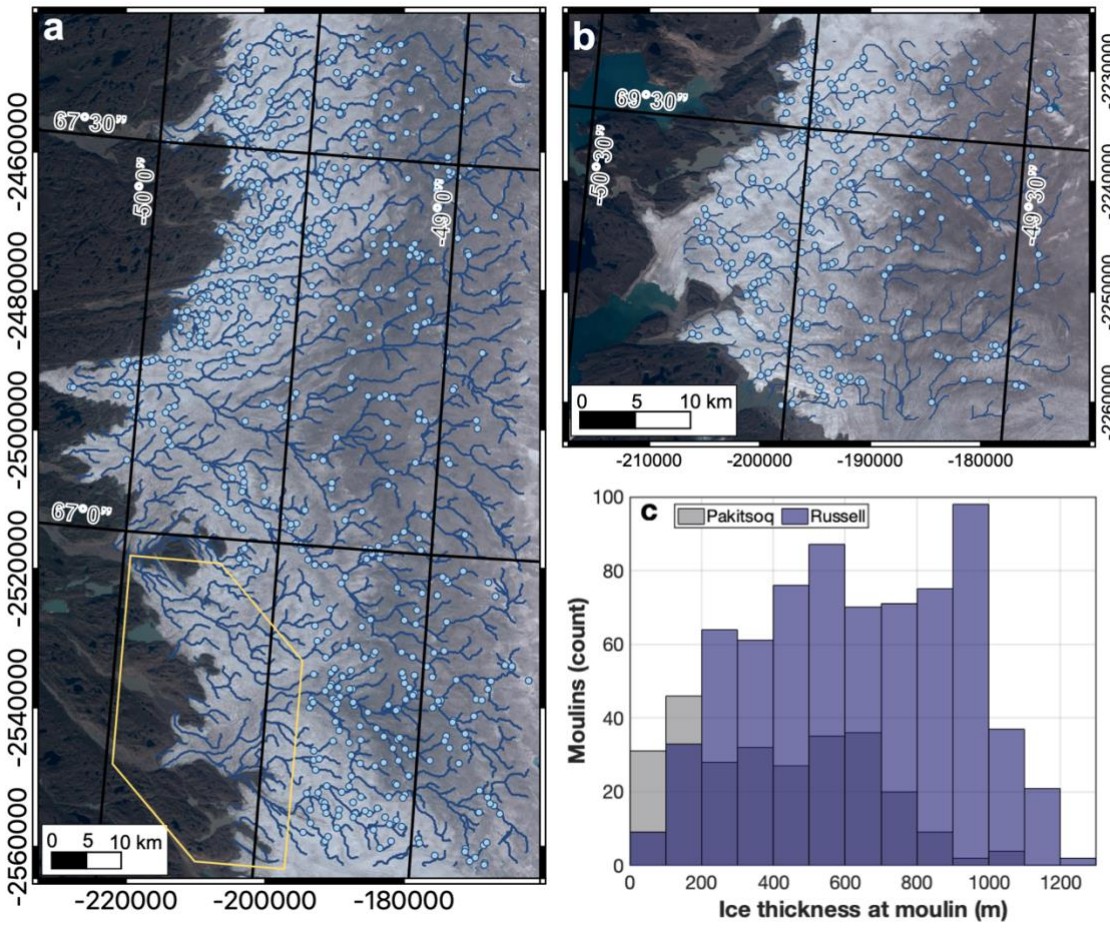

**Figure 3. Remotely sensed moulin locations and modeled subglacial flow pathways.** Subglacial flow paths predicted by Eq. (1) for (a) Russell glacier region (Smith et al., 2015) and (b) Pâkitsoq (Andrews, 2015; Hoffman et al., 2018) regions and (c) BedMachine (Morlighem et al., 2017, 2018) ice thickness at moulin locations. Background for a–b is the same as in Fig. 2a. Yellow outline in (a) is not included in subglacial pathway length calculations for Russell due to lack of surface imagery and moulin identification. Moulins occupy between 10% (Russell) and 14% (Pâkitsoq) of the englacial-subglacial efficient pathways below 1300 m elevation, suggesting that moulin geometry and evolution may be important to subglacial processes.



**Figure 4. Results of parameter sensitivity studies for 10-day MouSh model runs.** Shown are the sensitivity of moulin size to initial condition for moulin radius (a–b), enhancement factor for englacial ice (c–d), ice temperature scenario (e–f), Young's modulus (g–h), softness of basal ice (i–j), friction factor for water flow beneath the water line (k–l), friction factor for water flow above the water line (m–n), and ice thickness (o–p). The left column shows the moulin radii (black and grey) at the equilibrium water level and the equilibrium subglacial channel radius (purple) averaged over the final 24-hour period of the ten-day model run. The right column shows the equilibrium water level (blue), moulin volume (red), and volume of water in the moulin (gold) averaged over the same 24-hour period. Overall, moulin radius is most sensitive to the friction factors, while moulin water level and volume are most sensitive to ice thickness $H$ and basal ice softness $A$.




**Figure 5. Diurnal variations in moulin sizes in 10-day parameter sensitivity runs.** Shown are the sensitivity of diurnal variation in moulin size and water storage metrics to initial condition for moulin radius (a–b), enhancement factor for englacial ice (c–d), ice temperature scenario from coldest to warmest ice (e–f), Young's modulus (g–h), softness of basal ice (i–j), friction factor for water flow beneath the water line (k–l), friction factor for water flow above the water line (m–n), and ice thickness (o–p). The left column shows diurnal variations in moulin radii (black and grey) at the equilibrium water level and the subglacial channel radius (purple) in the final 24-hour period of the ten-day model run. The right column shows the diurnal variation in water level (blue), moulin volume (red), and volume of water in the moulin (gold) within the same 24-hour period.



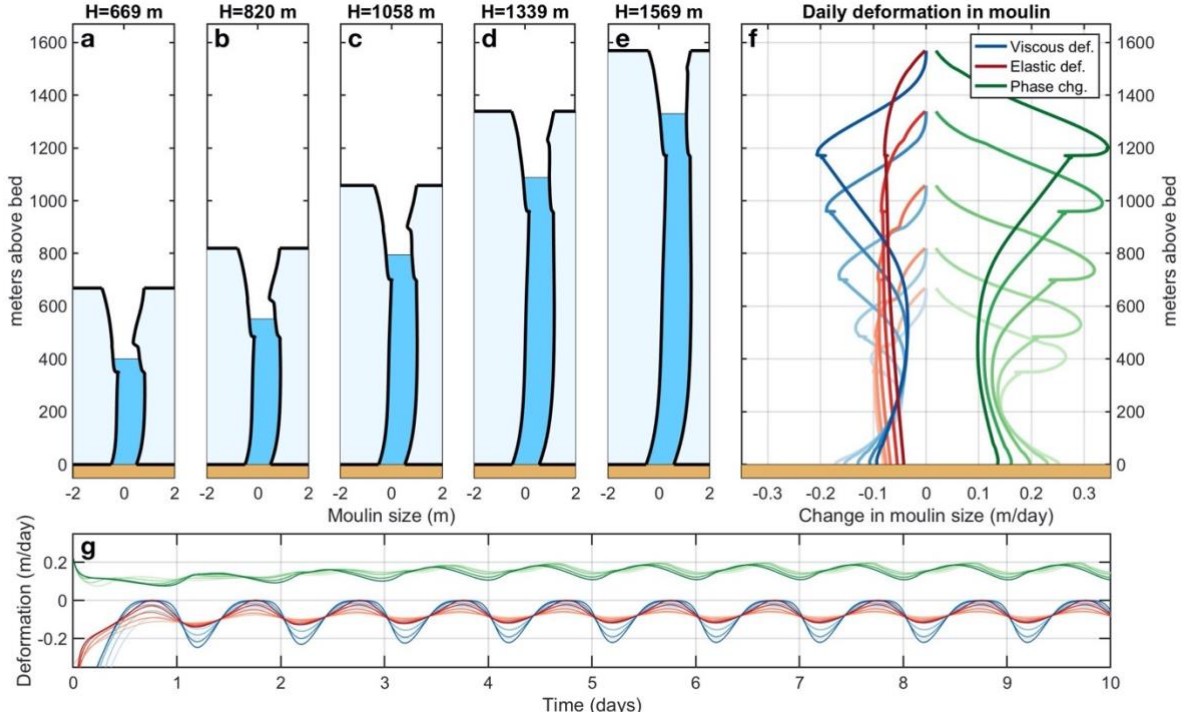

**Figure 6. Contributions of viscous deformation, elastic deformation, and phase changes to moulin geometry.** (a–e) Equilibrium geometries of five moulins in ice of different ice thicknesses $H$ (same as Fig. 6o–p) averaged over the final 24-hour period of a 10-day model run. (f) Vertical variation of viscous deformation (blue), elastic deformation (red), and phase change (green) contributions to moulin geometry averaged over the same 24-hour period. Negative values indicate contributions to moulin closure; positive values open the moulin. Darkening shades of each color map to moulins of increasing ice thickness. Closure and opening rates are greatest at the minimum daily water level (which is inferable by the lower notch in the moulin wall). (g) Time series of the components shown in panel f (colors the same) at the mean water level over the entire ten-day model run. The greater diurnal range in water level in moulins in thick ice drives the observed larger diurnal variations in viscous and elastic deformation.
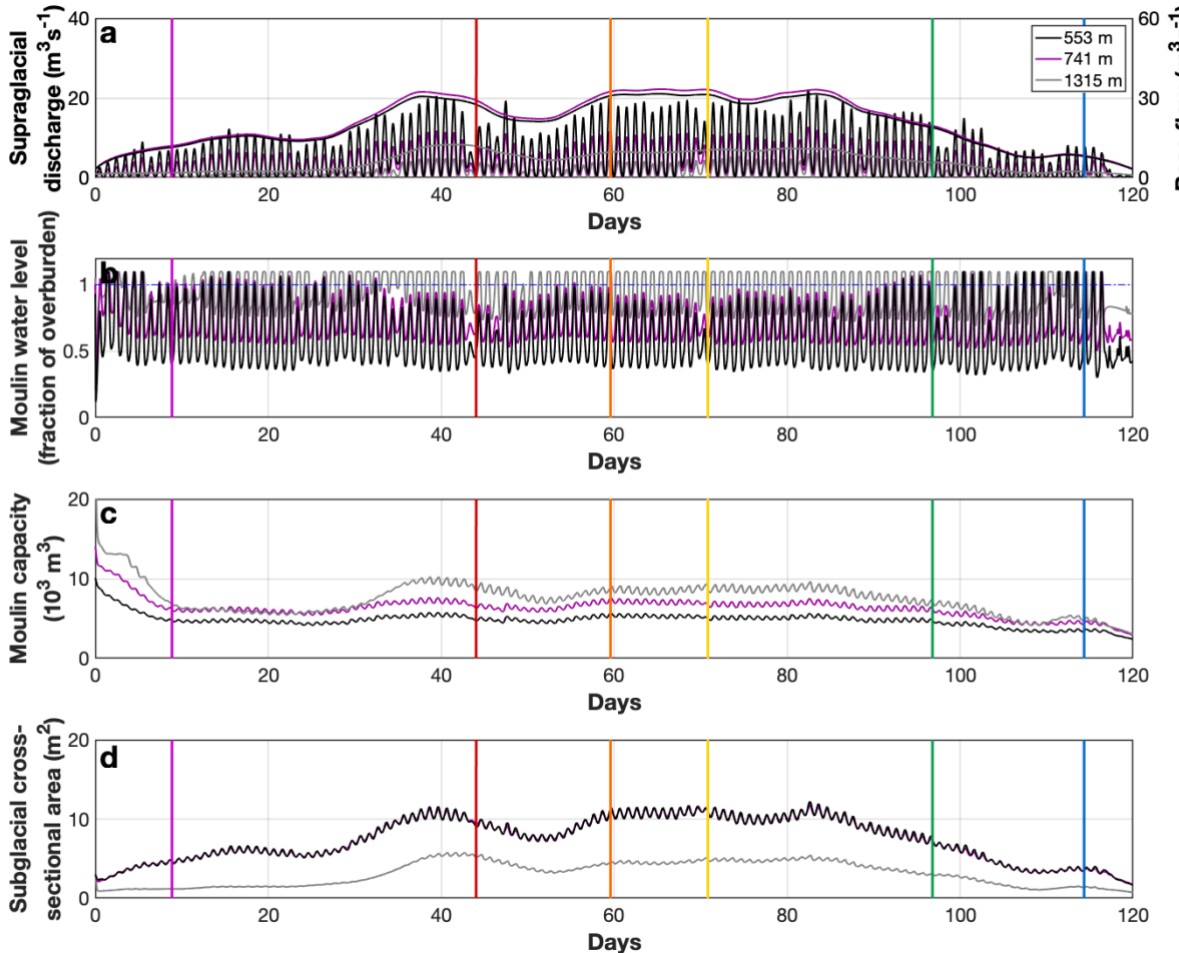

Figure 7. MouSh model runs with realistic supraglacial and ice conditions for a low-elevation basin (553 m ice thickness; black lines), mid-elevation basin (741 m ice thickness; purple lines), and high-elevation basin (1315 m ice thickness; grey lines). (a) Supraglacial discharge into the moulin and prescribed base flow. (b) Moulin water level as a fraction of overburden. (c) Moulin capacity, or the total moulin volume. (d) Subglacial channel cross-sectional area. Colored vertical lines indicate times in Fig. 8.

1140







**Figure 8. Evolution of moulin geometry over the melt season.** Colored boxes correspond to the times indicated in Fig. 7. (a–f) Basin 1 with ice thickness of 553 m. (g–l) Basin 2 with ice thickness of 741 m. (m–r) Basin 3 with ice thickness of 1315 m. Axes are not to scale.





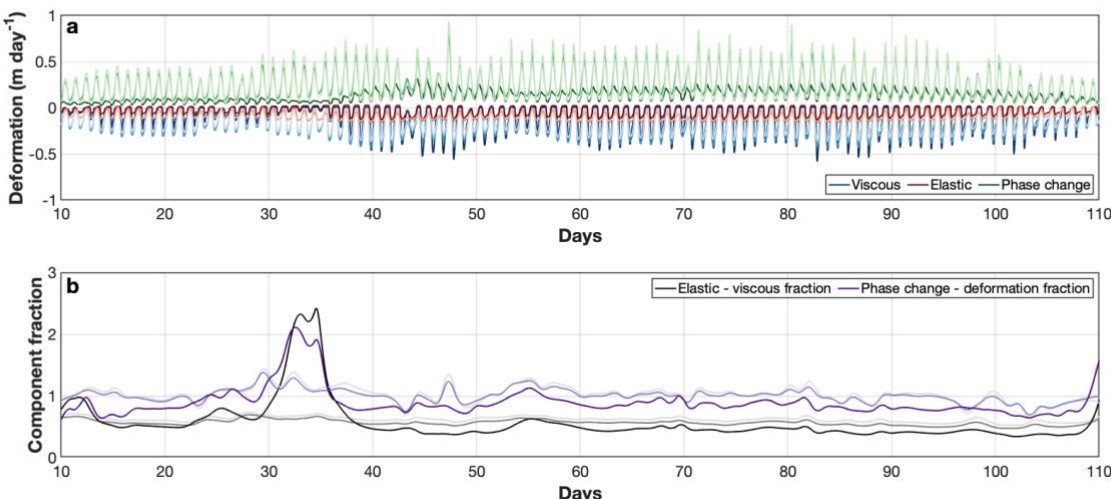

1145

**Figure 9. Time series of viscous, elastic and phase change components of moulin evolution and their relative importance in determining moulin geometry.** (a) Time varying viscous (blues), elastic (reds), and phase change (melting, greens) components of moulin geometry. (b) The ratio of elastic to viscous deformation (greys) indicates the relative importance of the two deformational processes in moulin evolution. All values are lower than 1, indicating that viscous deformation is always greater. The ratio of the total amount of phase change (melting above and below the water line) to total deformation (elastic plus viscous) (purples). Values above 1 indicate that melting dominates; values below 1 indicate that deformation dominates. Data is smoothed over 24h. For both panels, light colors are for Basin 1 ($H$=553 m), medium colors for Basin 2 ($H$=741 m), and dark colors for Basin 3 ($H$=1315 m).





**Figure 10. Daily percentage change in moulin variables relative to the daily mean value.** (a) Daily percentage change in moulin water level relative to the daily mean water level for Basins 1, 2, and 3 (black, purple, and grey lines, respectively). (b) Daily percentage change in moulin capacity relative to the daily mean moulin capacity. (c) Daily percentage change in the subglacial channel cross-sectional area relative to the daily mean value. For (b–c), colors are as in (a).

1160

**Figure 11. Difference between variable and fixed moulin geometries for Basin 1 and 2** (ice thicknesses of 553 m and 741 m, respectively). The fixed moulins are cylinders with a fixed radius of 1.4 m (Basin 1) and 2 m (Basin 2), which are the time-mean radius at the equilibrium water level for the variable moulins. In all instances, the difference is calculated as (*variable – cylindrical*) with instances of percentage difference calculated as 2(*variable – cylindrical*) / (*variable + cylindrical*). (a) Difference in moulin water level for Basin 1
1165  (black) and Basin 2 (purple). Negative values indicate periods where the variable moulin water levels are lower than those of the fixed cylindrical moulin. (b) Percentage difference in moulin capacity. When values are negative, the variable moulin is smaller than the fixed cylindrical moulin. (c) Percentage difference in subglacial channel cross-sectional area. These values are persistently positive, indicating that the subglacial channel is larger with a variable moulin. (d) Percentage difference in meltwater input. This value is always positive because the addition of water from turbulent melting increases the total amount of water in the system.





 **Table 1. MouSh model constants and parameter ranges.** During realistic runs (Section 2.4) Median values were generally used. In instances where values used differ from the median value, the values used is indicated in parentheses.

| Constant | | Value | | Units |
|---|---|---|---|---|
| $\rho_i$ | Ice density | 910 | | kg m$^{-3}$ |
| $\rho_w$ | Water density | 1000 | | kg m$^{-3}$ |
| $g$ | Gravitational acceleration | 9.81 | | m s$^{-2}$ |
| $L_f$ | Latent heat of fusion | 335000 | | J kg$^{-1}$ |
| $M_u$ | Dynamic viscosity (liquid water) | 0.0017916 | | Pa s |
| $K_w$ | Thermal conductivity (liquid water) | 0.555 | | J (m K s)$^{-1}$ |
| $C_w$ | Heat capacity (liquid water) | 4210 | | J (K kg)$^{-1}$ |
| $C_p$ | Heat capacity (ice) | 2115 | | J (K kg)$^{-1}$ |
| Parameter | | Median value | Range | Units |
| R$_0$ | Initial moulin radius | 2.4 (3) | 0.5 to 5 | m |
| E | Ice deformation enhancement factor | 5 | 1 to 9 | - |
| T(z) | Ice temperature | -6 (FOXX profile) | -23 to 0 | °C |
| Y | Young's modulus | 5 (9) | 1 to 9 | GPa |
| A | Basal ice softness | 6 x 10$^{-24}$ | 5 x 10$^{-25}$ to 5 x 10$^{-23}$ | Pa$^{-3}$ s$^{-1}$ |
| f$_M$ | Friction factor (under water) | 0.1 | 0.01 to 1 | - |
| f$_{OC}$ | Friction factor (subaerial / open channel) | 1 (0.8) | 0.01 to 1 | - |
| H | Ice thickness | 1058 (553, 741, 1315) | 669 to 1569 | m |





**Table 2. General ice and moulin input parameters for realistic runs**

| *Parameter* | *Basin 1* | *Basin 2* | *Basin 3* |
|---|---|---|---|
| Ice thickness (m) | 553 | 741 | 1315 |
| Distance from terminus (km) | 13.6 | 24.5 | 77.1 |
| Catchment size (km$^2$) | 19.8 | 18.4 | 55.5 |
| Moulin input, mean diurnal range (m·s$^{-3}$) | 11.5 | 6.7 | 2.5 |
| Moulin input, maximum value (m·s$^{-3}$) | 22.1 | 12.8 | 6.3 |
| Baseflow, mean value (m·s$^{-3}$) | 20.2 | 21.2 | 6.2 |

1175