# Peer review of "Controls on Greenland moulin geometry and evolution from the"

_The Cryosphere, 2021_

## Author Comment (AC1)

The paper is nicely written and details many aspects of the model which has a lot of components. Looking at the results for the viscous deformation, turbulent melting and the application of Glen's Flow law provides interesting information about the evolution of these features and how they might significantly change radius even on a daily basis, which is not something that I was aware could happen.

I do have some major concerns which I list below but, once these are taken into account, I think this will be a strong addition to glaciological modeling.

Thanks for the encouraging assessment of the worth of the MouSh model and our manuscript. We appreciate the time it takes to complete a thorough review. Our responses to both major and minor comments are below. We note that during the review process, many of the line numbers in the "line-by-line comments" section reverted to an automatic numbering scheme. We did our best to identify the correct lines and respond appropriately, but there are some instances where we were not successful.

**Major issues**

1. I'm confused by the implementation of elastic equations in this context. Most applications of elastic equations in glaciology, that I'm familiar with, are for a situation with a bending beam or plate (e.g. ice shelf flexure) in response to a changing force. I have a difficult time understanding how this can apply from constant ice force inwards into a moulin in an elastic rather than viscous form, particularly since elastic deformation should be instantaneous with changing force but here it is from the change in resistance (the water), and I'm not convinced that they are equivalent. As far as I know, this type of calculation with both elastic and viscous deformation in a moulin or borehole is new to glaciology and so the approach needs more explanation / justification.

The Reviewer is correct that the elastic deformation is instantaneous and thus only evolves with a changing forcing -- on ice shelves and other bending-beam glaciologic applications this is tidal forcing; in the case of a moulin, it is continuous variations in water level that change the stress state. The Reviewer is mistaken, however, that the elastic deformation results "from constant ice force inwards".  The cryostatic stress (constant ice force inwards, Equation 6a) is indeed unchanging, but the counteracting hydrostatic stress (variable water force outwards, Equation 6b) varies by the minute (e.g. Figure 7a) and thus drives continuous elastic deformation (as well as viscous deformation).

Visco-elastic deformation of a borehole/moulin is indeed new in glaciology, but it has precedent and builds upon earlier glaciology work by Weertman (1971, 1973, 1996), as well as rock mechanics work (e.g., Amadei, 1983; Goodman, 1989; Priest, 1993; and especially Aadnøy, 1987) that study elastic deformation in wells drilled for exploration geology.  Weertman, a solid-state physicist, applied dislocation-based fracture mechanics principles to vertically oriented geologic features that deform elastically: water-filled crevasses (downward propagating) and buoyant fluid-filled dikes in the mantle (upward propagating).  This work earned him the IGS Seligman Crystal (1983).  His equations for elastically driven crack propagation have since been applied to rapid drainage of supraglacial lakes (Krawczynski et al., 2009) and slower drainage of through-propagating hydrofractures (Poinar et al., 2017).  The geometry of the crack-propagation problem, however, is Cartesian (2D in x and z), which differs from the cylindrical symmetry of a moulin or borehole (2D in r and z).  For this reason, we drew on the rock mechanics literature (Aadnoy, 1987), where the geometry is cylindrical.  We added clarifying sentences to Section 2.2.2.1 to describe the application from rock mechanics to glaciology.

In addition, we discuss "the role of elastic deformation in ice sheet hydrology" in Section 4.5, which we retitled accordingly. Based on our findings with the MouSh model, we suggest that the role of elastic deformation in hydrology models that resolve sub-daily variations should be reevaluated.

Part of the problem is that equation 4 is difficult to follow in its current form. If this is a new way to apply elasticity to a moulin then the equation needs to be fully derived in an appendix. If it has been applied in glaciology before you need citations. You base your equations from Aadnoy (1987), but this isn't even included in the reference list.

We see now that we described the elastic deformation component of MouSh too briefly in the manuscript.  A detailed but relatively informal description of the derivation of Equation 4 for elastic deformation is currently available in our Github repository (github.com/kpoinar/moulin-physical-model/releases).  We formalize this document as a chapter in a Supplement and reference it in Section 2.2.2.1.  Finally, we have added Aadnøy (1987) to the References -- we regret the mistaken omission.

Both the elastic and viscous deformation rates that you plot are higher than I would have assumed for this situation. Previous analysis on borehole deformation has closure rates one or two orders of magnitude smaller (e.g. Paterson, 1977). Also Catania and Newmann (2010) argued closure would primarily occur in the base of the moulin and not the top 80%. A discussion on why yours are so much higher and/or examples of other systems that deform this rapidly would help.

Comparison and additional discussion of our deformation rates relative to previous results is a good idea. We now include a section in the discussion focusing on comparison with previous model and observational results ("Relative importance and magnitude of moulin deformation").

The reason that moulin closure rates are much higher than closure rates in boreholes in the ice-sheet interior (Paterson, 1977) is simply water pressure: the moulin water level is nearly always below flotation (rough range 0.55-0.9 of overburden; Figures 4-5), whilst boreholes are generally kept filled to flotation with drilling fluid. This is the same reason that viscous closure rates are greatest near the water line (blue lines in Figure 6f): the inward cryostatic stress is least offset by the outward hydrostatic stress there.

Catania and Neumann (2010) actually model the closure of an air-filled moulin (see paragraph 10 of that paper). In this instance, there is no oppositional hydrostatic pressure; therefore, the rates of closure will be highest at the bed. We model moulins that contain water, which resists creep closure (Equation 6) and results in the highest creep closure rates to occur at the water level. In general, our creep closure rates are similar to those of Catania and Neumann (2010) near the surface, where our moulin is also air-filled, but lower near the bed, where our moulin is water-filled. A detailed comparison is now included in the new discussion section.

We know of no other vertically oriented glaciologic systems that are forced to deform as rapidly as hydraulically connected moulins. Furthermore, field observations of the sizes and deformation rates of moulins or boreholes are scarce. Measurements of moulin geometry are limited to the end of the melt season, when the absence of aggressive diurnal hydrologic forcing allows cavers to explore them with relative safety (Covington et al., 2020); this precludes observation of diurnal deformation rates.

2. Following on from this, where have you got your Maxwell times of 10-100 hours from? I believe the Maxwell time should be more in the range of a few hours. Therefore viscous deformation should be the primary application for moulin shape evolution. I'm also very unclear how you're transitioning between elastic and viscous deformation with this model and applying the Maxwell time. It seems these are both being calculated separately but continuously given that you are plotting both over sub-hourly, multi-day timescales?

In the manuscript, we explain that "The Maxwell timescale is equal to $(Y \times A \times \tau^2)^{-1}$, or roughly 10–100 hours for typical Greenland ice" (Lines 115-116), which comes from the Turcotte & Schubert (2002) textbook (Section 7.10 Viscoelasticity, Equation 7.239, page 376) with adaptation to glacier viscosity $1/(A \times \tau^{n-1})$ with n=3. (The Turcotte & Schubert formula has an additional factor of 2 that comes from the two shear stress terms in the expression for deviatoric stress in a 3D fluid; we dropped this inadvertently, but we model a 2D system, where deviatoric stress would have only one term, not two, so our expression is accurate.)

We obtained a numeric value for the Maxwell timescale using the values in Table 1: Young's modulus Y~1–9 GPa, ice softness (flow law parameter) A~$5\times10^{-25}$ to $5\times10^{-23}$ Pa$^3$s$^{-1}$, and typical shear stress ~10–100 kPa. This gives 20,000– 200,000 seconds, or 6–56 hours. We rounded this to an order of magnitude, 10–100 hours.

A Maxwell model, or an elastic element (a spring) and a viscous element (a dashpot) in series), is the simplest form for a visco-elastic system, and a standard in geophysical modeling. A Maxwell model adds both components of deformation in a simultaneous and linearly independent way. This is consistent with the Reviewer's description of calculating both components "separately but continuously". We add an explanation of this at the end of the relevant paragraph in Section 2.2.2.

Vaughan (1995), in his well-known survey of tidal flexure on ice shelves, finds shear modulus μ ~ 1 GPa. Snow and firn cover these ice shelves; their lower value of shear moduli compared to ice likely biases these measurements low. For lake-driven hydrofractures in western Greenland, Krawczynski et al. (2009) used a range of μ ~ 0.3–4 GPa, reflecting pure ice (no firn cover). For subsurface hydrofractures in Southeast Greenland, Poinar et al. (2017) found a likely shear modulus μ ~ 0.2– 1.5 GPa. Our sensitivity test values, Young's modulus Y ~ 1–9 GPa (or, equivalently, shear modulus μ ~ 0.3–3.5 GPa, using Poisson's ratio ν ~ 0.3–0.5), are the full likely range for glacier ice. We add a paragraph to the Supplement to explain our choice of Young's modulus values.

3. The treatment of turbulent melting and refreezing is confusing. Why only include refreezing outside of the melt season? Do you assume no refreezing overnight during the melt season? I see you say refreezing occurs only when water flow is laminar but it's not clear to me that water flow will always be turbulent from the beginning to the end of the melt season. This needs more justification in the text by reporting the expected Reynolds numbers.

The refreezing module activates when water flow does not occur (a static water column) or when water flow is laminar (no heat is generated due to the dissipation of turbulent energy). These rules for refreezing use a calculated Reynold's number to determine the flow regime throughout the moulin (laminar flow when Re <2300).

During the melt season, the meltwater supply to moulins, even during the evening, is substantial due to supraglacial basin size and hydraulics (e.g. Smith et al., 2017). All moulins we model have at least a modest flow in the early morning (time with the lowest flow, now included in Table 2).

These requirements and the parameters of our chosen representative model runs mean that refreezing is not activated, except near the end of the melt season. We now include the range of Reynolds numbers for the model runs in Tables 1 & 2. We also more clearly describe our thinking and choices in Section 2.2.3.1.

4. I understand why you've applied a simplified subglacial model given the complexity of the moulin model. However, both the description of the basal channel model and the application are confusing.

From what I can gather you're calculating channel characteristics at the moulin outlet (using ice pressure and moulin head pressure) but are applying a constant hydraulic potential gradient from the moulin to the terminus, so only producing one output point. The length scale calculations from moulin to terminus are not ideal in application to a continuously evolving channel (which will not be linear in terms of pressure) and are likely unrealistic. Instead why not apply a range of hydraulic potential gradients to test how those impact the moulin evolution? That would be much clearer to show how the pressure change at the bed impacts and is impacted by the moulin head.

The subglacial module uses the time-varying gradient between the water level in the moulin and atmospheric pressure at the terminus. Thus, the channel size and rate of change relevant to MouSh are at the midpoint between the moulin and terminus. These are the only two hydraulic heads that can be reasonably inferred without using a substantially more complex subglacial hydrology model that would shift emphasis away from the moulin and onto the subglacial system. However, we do more clearly describe the outputs from the subglacial model and how the moulin is connected in Section 2.2.4.2.

The Reviewer rightly points out that a sensitivity study of the effect of the hydraulic gradient would be relevant. During our sensitivity tests, the hydraulic gradient is varied as a result of differing ice thicknesses and distances from the terminus, albeit not in isolation (Figures 4o,p & 5o,p). We now include lines discussing the impact of the hydraulic gradient, including a fourth realistic model run, in Section 3.3 and the Supplement.

One of the significant concerns I have about the channel is the necessity of a large base flow. Looking at Figure 7 the base flow is the main driver for channel evolution and at an input rate of ~20m3/s that's not surprising it's the primary control. To better determine the role of the changing moulin head it would be better to avoid adding additional time-varying water inputs at the bed since it's not clear that it's at all realistic. Instead, a static background water flux and/or a larger initial channel size could help with stability issues.

Indeed, the baseflow element required by the current setup of the subglacial model (a single channel connecting the moulin to the ocean) is non-ideal, but it emerges as a natural consequence of the simplicity of our subglacial model. The baseflow parameter represents all subglacial flow in the vicinity of the moulin (including all flow upstream and any nearby flow in adjacent branches of the dendritic network) except for its direct outflow, which is contained in the single-channel model. While 20 $m^3$/s baseflow does seem large compared to supraglacial inputs ~10 $m^3$/s (Figure 7a), in the context that baseflow represents all accumulated inputs from all nearby moulins, it is more reasonable. However, we do acknowledge that the values are large. In order to mitigate these large values we have made several changes to the model initial conditions and characteristics (reflected in Figure and Table and textual changes throughout the text). We also perform a comparison of the impact of baseflow, with two new model runs, in the Supplement.

However, we choose to maintain the seasonal signal in the baseflow because the characteristics of the seasonal cycle are well known both from supraglacial and proglacial studies.We also note that our sensitivity study does not use a baseflow or background flux.

It's generally hard to believe the channel outputs that you present as it's not clear what the differences are between basins and experiments in terms of the hydraulic potential gradient, and because of that large base flow rate.

We make an effort to clarify the characteristics of the realistic basin characteristics, including hydraulic gradients, in Table 2 and in Section 2.4. We also made modifications to the baseflow (see previous comment and response).

However, as this paper focuses on the moulin model, so should the results and discussion. The role of a basal channel in this case is to present semi-realistic evolution characteristics to feedback with the moulin water levels. This does not give you much information about what is happening at the bed anywhere downstream of the moulin so that should not be widely discussed. Along these lines, before you begin your moulin model methods, you look at subglacial channel routing in section 2.1, which is misleading for the reader. This section does not seem relevant to this paper because of the highly simplified nature of the channel model that you apply and it would be better to start the paper with the moulin model methods.

We agree with the Reviewer. The primary purpose of this manuscript is to describe the model of moulin geometry and not discuss the subglacial system. We include discussion of the subglacial system in two locations within the manuscript. First, briefly in the description of model inputs, and finally in the discussion when we elaborate on the potential of coupling the MouSh model with a more complex subglacial model in order to eliminate the need for our parameterized base flow and what processes may be lumped in our simple tuning parameter. In this section, we emphasize the simplicity of the subglacial model and the benefits a more realistic subglacial model would provide. We do not attempt to interpret what may be going within the subglacial system downstream. The results and discussion *do largely* focus on the moulin, rather than the subglacial system.

In recognition of this comment, we've shortened the discussion Section 4.6. We have also entirely removed the small remote sensing component (Sections 2.1, 3.1, and parts of 4.1) in order to eliminate any confusion or implication of interpreting basal conditions. This removal results in textual modifications throughout the text.

5. The discussion at the moment focuses a lot on how moulins are formed, the subglacial system, and englacial void ratios. These don't seem directly relevant to your main findings from this complex model, which are the changes in shape, melt rate and deformation of the moulin. Particularly given that the subglacial model is much more simple and this is the first step in coupling to a more dynamic subglacial model, the discussion in this paper should be focused on the moulin evolution. There are many interesting outputs from your model runs that you could discuss in terms of the deformation of the ice possibly moving the input of the moulin at the bed along with stretching the length of the moulin; where in the moulin and at what time of the season water would be stored at higher or lower pressure influencing the subglacial system; the influence of the moulin shape on the head etc.

We thank the Reviewer for the three interesting ideas for new model experiments. We don't see a way to adapt these ideas as *Discussion* topics; rather, these new experiments would provide additional *Results*. We agree that they would likely be interesting, but they are outside the scope of the experiments we did perform, and thus would merit a different paper.

We will address the Reviewer's general criticism that our Discussion section extends too far from the direct results as follows:

- We will retitle Discussion Section 4.1 (currently "4.1 Formation mechanisms of moulins") to "4.1 Timescales of moulin formation and evolution", which we can directly comment on using the literature (formation timescale) and MouSh (evolution timescale). We also shorten this section (removing two of its four paragraphs) to sharpen its focus to the implications of our MouSh results.

- We defend Discussion Section 4.4, "Moulin geometry and the englacial void ratio", as it extends beyond our direct results while still being fundamentally grounded in them. This section cites multiple of our figures (Figures 4, 7, 8, 10) alongside current ideas in the literature, notably brought up by Flowers (2015). It puts our direct results, which as the Reviewer points out are "the changes in shape, melt rate and deformation of the moulin", into the context of an adjacent idea (the englacial void ratio). This is a natural extension to a related and current topic that is entirely appropriate for a discussion section.

- Discussion Section 4.6, "Potential coupled englacial – subglacial hydrology models", essentially addresses the main weakness in our model setup: the simplistic subglacial hydrology model we use. The Reviewer shares our awareness of this point, as evidenced throughout this review. To emphasize that this section is a critique of our own work, rather than an extension or a look toward the future, we will retitle it "Limitations of the current MouSh englacial – subglacial hydrology model".

**Line-by-line comments**

11. where does that 10-14% number come from?

This section has been removed to focus the manuscript more clearly on moulin geometry.

12. they constitute most of the englacial system – what about englacial channels?

For our Greenland study, we treat "englacial channels" and "moulins" essentially as synonyms: near-vertical conduits to the bed. On other glaciers (especially, for instance, temperate debris-covered glaciers in the Himalaya), englacial channels formed by cut-and-closure can be quite sub-vertical (Gulley et al., 2009) and, as the Reviewer implies, can thus constitute a significant fraction of the englacial-subglacial drainage system. We address this in the shortened Section 4.1.

38-39. you already said this in your first sentence of the introduction.

Good point; we will remove this sentence.

71. what do you mean 'relative path length'? This whole paragraph is confusing because you're discussing basal hydraulic potential rather than moulins.

This section has been removed to focus the manuscript more clearly on moulin geometry.

84-95. I'm very confused. what do you mean by theoretical flow accumulation? Are you saying you're defining the catchments at the base of the ice? You're defining channel lengths at the bed? What is a subglacial channel node? Channels should join up dendritically towards the terminus in any case and are therefore linked rather than in separate segments.

This section has been removed to focus the manuscript more clearly on moulin geometry.

104. why do you initiate with a semi-circular, semi-elliptical shape? There doesn't seem to be any reason for this and it primarily serves to complicate your equations and your analysis.

The circular half of the moulin is the downstream end, where a simple circular geometry (radius r1) is an accurate description of the shape supplied by visco-elastic deformation. On the upstream half of the moulin, incision by the inflow stream and falling water greatly increases the radius of the moulin; hence, we describe this as an ellipse, with a long axis (r2) and a short axis (r1). We added an explanation for our initialization shape and a reference to Figure 2 in Section 2.1.1.

116. undefined parameters for Maxwell time. Where do you get the equation in parenthesis from?

We added parameter definitions and repeated the citation of the Turcotte & Schubert textbook in Section 2.2.2. See our response to the related Major Comment #2, above.

136. if moulins form by drainage into crevasses and hydrofracture why do you assume it's compressive?

While some moulins are associated with crevasse fields and persistent extension (e.g. Smith et al., 2015, Figure S2), moulins at moderate and high elevations are generally associated with supraglacial lake drainages (e.g. Figure 2; Poinar and Andrews 2021; Stevens et al., 2015; Hoffman et al., 2018). In these locations, ice is persistently compressive, except during a short period of extension during the initial formation. We include a reference to Poinar and Andrews (2021) to clarify this point.

146. you say viscous deformation is the dominant process over a 1 day timescale but you plot your viscous deformation on much smaller timescales showing diurnal variation.

Excellent point. We will remove this sentence.

154-156. specify here this opening and closure is relative to the pressure difference at depth – moulin should not open at all depths when above flotation – only in regions of the moulin where the relative pressure is higher than flotation. Looks like you're calculating this in the next section but this should be clarified here.

Good catch. It now reads: "When water is above flotation, the moulin will viscously open in regions where the hydrostatic pressure exceeds the cryostatic pressure."

160. why have both equation 5 and equation 6?

Good point, we now combine these equations.

189. laminar flow is when the Reynold's number is less than 2300.

This has been fixed here and discussed in a new Supplemental section.

193. do you mean all ice is at the melting point, not water?

No, we meant all water is 0°C (or, if at depth, it's at the pressure melting point).  We changed our wording to "freezing point" to try to make this clearer.

206. what difference do you find with these alternative approaches?

Ultimately, we prefer using the first refreezing parameterization (Poinar et al., 2016). We removed reference to the second refreezing module.

210. you haven't told us about the equation you use for turbulent melt. I see that you have it later in the paragraph but it's confusing in this line because it implies we already know how you calculate it.

We rewrote this section to better reflect the previous section and to improve clarity.

226. but you do assume it's at the pressure melting temperature in your refreezing section. I'm getting confused.

Yes, see previous response (line 193).  All water in the moulin is at the pressure melting point.

251. how modest?

Varying the friction factors across an order of magnitude does not affect the equilibrium  or diurnal variation in water level; however, increasing friction factors by two orders of magnitude increases the equilibrium and diurnal variation in moulin water storage by about 10-20%.  We now include this number and a reference to Figure 4.

266. what is S in equation 18? Again I thought the ice on the moulin wall was at the pressure melting point?

No, see previous response (line 193).  We use measured ice temperatures for the ice surrounding the moulin; these can typically be -5°C to -15°C, depending on depth (Lüthi et al., 2015).  We did test one scenario with moulin wall ice at 0°C, but more generally, MouSh allows ice temperature to vary.

S is the cross-sectional area of the subglacial channel. We now define the previously undefined variables following the equation.

318. presumably the unit hydrograph is to allow a lag for the runoff to reach the moulin? If so, you should state that.

We now include the sentence: "The use of a unit hydrograph parameterizes the impact of ice surface routing on the moulin input hydrograph."

322-329. this last paragraph seems more appropriate in the next section

Good point. We have made this change.

325. specify what you mean by englacial void ratio here? Why would that impact flow from upstream?

We clarify this sentence: "This englacial storage is often represented as an 'englacial void ratio' in current subglacial models."

332. what two elements?

Moulin and subglacial channel. We revised this sentence to read: "...that simulate flows between the moulin and subglacial channel"

339. assuming b is elevation above sea level, why include it if you have zero bed slope?

We include b for completeness and because the model has been developed to run with a non-zero bed slope. We now define b.

358. where is the base flow added?

We add base flow, in addition to discharge from the moulin, to the subglacial module. We add a clarifying sentence here.

359. you mentioned Qin above in section 2.2.4.1, which included the baseflow. That's not being added directly into the moulin I assume? This should be clarified.

See response to the above comment.

371. You need a justification for your choice of enhancement factor. It seems like you're applying this factor between 1 and 9, but measurements by Luthi et al (2002) in Greenland ice suggested it can reach up to 2.5 in Holocene ice but is closer to 1 above that depth.

We choose an order of magnitude (1-9) to match the range of other rheological parameters (Young's modulus, ice column flow law parameter A via ice temperature scenario). We agree this is larger than E is likely to vary in the field. We have added a description of our reasoning here (Line 372).

398. what are these basins?

Yikes, yes, this is an awkward first mention of the basins, which we neglected to define. We now include a sub-section describing how we defined and delineated the test basins.

409. why are these lengths so different? I don't think these lengths scales help your model application – it makes it confusing (see comment at the beginning).

This section has been removed to focus the manuscript more clearly on moulin geometry.

419. how can they reach equilibrium with constantly changing water input and a constantly evolving basal channel? Or is this with a constant input?

Yes, the key word here is "near" equilibrium (or quasi-equilibrium). The forcing (water input) and the internal oscillation of the system (basal channel and head) continually vary, causing the moulin size to vary likewise. We have added a figure to the supplement that illustrates this ~3–5-day relaxation time alongside the continual evolution of moulin size. We replaced "near equilibrium" with "quasi equilibrium", which is more accurate.

[Figure]

440-445. Changing Youngs modulus for elastic expansion increased moulin volume by 38% and capacity by 56%? This seems like a much more significant change than I would assume from elastic deformation in this context.

Yes, it was larger than we expected, too. We studied it closely and devoted an entire Discussion section to it (Section 4.5, "The role of elastic deformation in moulin geometry"). Ultimately, its effects are comparable to viscous deformation (which we also varied by an order of magnitude, via the enhancement factor E), which showed a 20–40% change in moulin geometry factors. These are comparable to the 38–56% changes sourced elastically, cited above.

461. it would be more appropriate here to say the outflow is sensitive to the steepness of the basal pressure gradient.

We do agree that the outflow is set by the hydraulic gradient, but in turn the hydraulic gradient is set by the distance from the terminus (L) and the ice thickness (H), as we've written, because ice thickness sets the rate of subglacial creep closure and is a ceiling for moulin water level. The basal ice softness ($A_{basal}$) also plays a key role in channel size and thus outflow -- high H and low $A_{basal}$ allow for higher moulin water levels and thus a steeper hydraulic gradient to the terminus. To incorporate this comment, we have included the line: "This sensitivity indicates an interplay among these model parameters, the subglacial hydraulic gradient, and moulin water level."

Line 474?. but the increase in water level should increase the pressure gradient and cause faster flow through the subglacial channel and melt opening?

We think we have this line number correct. If so, we refer the Reviewer to the previous comment.

Line 486?. this discussion and Figure 6 show a diurnal change in viscous deformation by up to 20cm and 10cm elastically. Then the diurnal phase change up to 30cm/day. Is that saying you melt up to an extra 30cm a day? And that the moulin diameter pulses in and out every day due to melt countered by viscous and elastic deformation? Is there any evidence for this from moulin measurements.

Unfortunately, we were unable to identify to which line the Reviewer is referring, but we believe that this comment refers to Section 3.2.2. Figure 6f plots the cumulative change for viscous deformation, elastic deformation, and melting. While we do calculate values of between 20-30 cm per day near the water line, when combined, the actual daily change in moulin diameter is smaller because the moulin is in quasi-equilibrium. For clarity, we now include the combined change in moulin shape in Figure 6g.

Yes, the Reviewer correctly states our results. Unfortunately, measurements of moulin geometry are not available until the end of the melt season, when the absence of aggressive diurnal hydrologic forcing allows cavers to explore them with relative safety (Covington et al., 2020). This precludes observation of diurnal deformation rates.

516. what would cause a moulin to change radius diurnally more in thicker vs. thinner ice?

This is the internal dynamics of the subglacial model (Schoof, 2010): daily water pressure fluctuations are greater in thicker ice. This is likely due to the nonlinearity in subglacial creep closure ($\sim H^3$) alongside linear channel opening by melt. The moulin water level drives visco-elastic opening and closure of the moulin, causing higher diurnal variability in moulin size in thick ice. We now add a sentence to this effect.

Line 535-536?. the similarity between Basin 1 and 2 is likely because the basal channel model is driven by a similarly large background influx rather than by the changing conditions of the moulin itself.

Despite similarity in supraglacial and baseflow inputs between Basin 1 & 2 (Figure 7a), there are clear differences in moulin water level and capacity. These differences are elucidated in Figure 7 and in the paragraph starting on line 530. So, some other driver (not water inputs) explains the differences in moulin capacity; this leaves ice thickness or basal hydraulic gradient. We remove the ending phrase of the sentence starting on line 535 to address this.

526. how could thick ice viscously close channels if water is above overburden pressure?

In Basins 1-2 (thinner ice, H < 750 m), water rarely exceeds the overburden pressure (Figure 7b). In Basin 3 (thicker ice, H>1300 m), as the Reviewer points out, the pressure exceeds overburden for a number of hours each day. These hours without visco-elastic closure should overall increase the mean channel size, compared to a moulin that did not overflow daily (e.g., one that had higher base flow). We now add this caveat to the discussion -- that closure rates in Basin 3 are retarded by the high daily water pressures.

Line ??.  specify which system sees the increase.

The line number that this refers to was garbled by the Reviewer's text editor. We were able to reconstruct line numbers for most comments, but not this one. Unfortunately, we cannot make the suggested change without knowing which part of the text the comment refers to.

Line ??.  you have elastic processes in the channel too? Any references to show this is justified in basal channels?

No, we do not have elastic deformation in the subglacial channel model. See the last paragraph in Discussion Section 4.5, for a discussion on how inclusion of elastic processes in subglacial channel models would likely affect subglacial model output in general.

573. I'm unsure why you're discussing initial moulin formation processes which aren't the focus of your study. Moulin evolution, yes, and you have plenty of interesting things to talk about on that subject.

See our response to an earlier, more general comment:  We will retitle Discussion Section 4.1 (currently "4.1 Formation mechanisms of moulins") to "4.1 Timescales of moulin formation and evolution", which we can directly comment on using the literature (formation timescale) and MouSh (evolution timescale).  We will also shorten this section (removing two of its four paragraphs) to sharpen its focus to the implications of our MouSh results.

640. exploration would be good to validate your model so I wouldn't discount it.

We agree that exploration in general would have utility.  We now soften the language here regarding the relative value of underwater versus sub-aerial exploration.

645. can you clarify this sentence. You are saying field measurements show 103-112% of overburden, or 3-12% of overburden? The former seems more likely. But 20% above overburden? That should be on the surface?

It's the latter, but I see how our sentence is misleading.  It's important to note that the mean daily water pressure in a moulin is generally *much* lower than overburden (<70% of overburden in Greenland observations to date), so even these larger fluctuations (25%) will not overflow a moulin. Observed diurnal water level fluctuation at one of the Covington et al. (2020) moulins was quite small: 3% of the ~640 m overburden, or just 20 m daily variation around a daily mean water level of ~460 m (~65% of overburden).  At the Andrews et al. (2014) moulins, daily variations approached 25% of overburden, but the daily mean water level was ~70% of overburden (~500 m in ice thickness 720 m), meaning that the moulin water level rarely exceeded overburden, and never overflowed. We now add text to remind the reader that the mean daily water level is actually quite low (<70% of overburden), not near flotation as some might assume.

646. some boreholes hit more efficient systems, as explored by Meirbachtol et al (2013).

This is true, which is why we use the phrase 'generally thought'. In Greenland, there are some instances of connecting to an efficient system with a borehole, but most connect to the inefficient system, including some described in Meierbachtol et al. (2013).

656. rephrase 'variations in diurnal water level variability'

Yup, that's awkward.  We will remove the word 'variability'.

669. I don't think an englacial void ratio is used to resolve diurnal basal pressure. How would you get a spatially variable englacial void ratio? What has this got to do with moulins?

The englacial void ratio is in fact introduced for this purpose.  All current models use a fixed (not spatially variable) value for englacial void ratio that is tuned to an order of magnitude that works (allows the model to accurately reproduce basal water pressure observations). We include a reference to Flowers & Clarke (2002) for clarity.

Line ??.  you said earlier in the paper that moulins are used as source inputs for models. How does this link to englacial void ratio? The change in water level is a moulin because of increase/decrease in diameter will impact the water supply to the base via the pressure. Perhaps you mean a storage parameter in models? I certainly think it's worth coupling with subglacial hydrology, but I'm not sure this paragraph makes sense. Line 696 covers this possibility and is an important point to make.

We hypothesize that (in the real world) moulins can both be source inputs and storage features.  In current subglacial models, they serve only as source inputs; we posit that moulins can also subsume the englacial void ratio / bulk storage parameters commonly used in subglacial models. We rewrite the sentence starting on line 668 to clarify how the englacial void ratio works in subglacial models. We also tweak the language in the following paragraph to better describe how including moulins with variable geometry could reduce subglacial model dependence on the englacial void ratio, which is the primary short-term storage parameter in subglacial models.

700. what do you mean a static shape instead of static cylinder? The Trunz et al, in review paper is mentioned a lot which is frustrating since we don't have access to see what it discusses.

Here we mean an arbitrary shape that doesn't change, versus a cylinder shape that doesn't change. We now add the word "arbitrary" for clarity.

731-739. see my above comments about elastic deformation. You need more justification for these statements given that it hasn't been included in subglacial models to date.

See our above response.  Our justification for suggesting inclusion of elastic deformation is found on lines 730-731: in a water-filled moulin, it is comparable in size to viscous deformation.

742. are you sure it's not that subglacial channels form where there are moulin inputs?

This is likely a chicken or egg problem (see, for a similar example, Sergienko, 2013). However, we removed the remote sensing part of the manuscript, including Figure 3, so this sentence has been removed.

Line ??.  rephrase this sentence.

We were unable to reconstruct line numbers for this comment.  Unfortunately, we cannot make the suggested change without knowing which part of the text the comment refers to.

**Figures**

Figures 4 and 5. Why does the y axis of the diurnal range go up to 0.4 if values don't go above 0.2?

Currently, the y-axis range in Figure 4 leaves room for the panel numbers. However, we do decrease the y-axis range slightly in Figure 5. In Figure 5, the values go up to 0.21 and room is needed for the panel letters, so the adjustment reflects that.

Figure 6.  I'm intrigued by the shapes in f. Why is there more turbulent melting in the middle of the borehole? What are the factors contributing to the differences between elastic and viscous deformation shapes and rates? In g since it seems to have reached equilibrium within a day or so it would be useful to zoom in so we can see the lines better.

We now add panel h, which shows a zoom (Day 9-10) of panel g.  We also now add a black line to panel f that shows the total mean daily deformation (viscous + elastic + phase), as mentioned in an earlier comment.

Figure 7. In your thickest ice example for moulin water level, it looks like your moulin is overflowing. Also in d) where is the 741m example? The channel size looks almost entirely dictated by the background flow you input with small diurnal variability on top.

The Basin 3 moulin does overtop due to the ice overburden pressure -- decreasing the base flow results in even more persistent overtopping. This is mentioned in Section 3.3 and again in 3.4. We now also indicate this in the caption. In panel d, the 741 m moulin (purple) is hidden behind the 533 m moulin (black).  We also clarify this in the legend.

Figure 8. This is a really interesting figure. Why not discuss the shape changes (particularly due to Glen's Flow law) more in the manuscript?

The bow shape from Glen's Flow Law has no effect on moulin volume (water storage) or water level. We find no importance to it, but we include it in the model for completeness: this is the true *shape* of a moulin, even if it has no effect on the dynamic behavior.

Figure 9 b. What happened around day 32?

During this period, the minimum daily supraglacial inputs are quite high (Figure 7a), this in turn results in water levels staying around flotation (Figure 7b). In this case, viscous deformation hovers around zero (though causing moulin opening), resulting in a high ratio of ecstatic to viscous deformation and a high ratio of phase change to viscous deformation (purple line in Figure 9b). There is an associated growth in moulin capacity (Figure 7c) Ultimately, this is a response to multiple days where melt inputs do not exhibit substantial diurnal variability.

**References (review + response from authors)**

Aadnøy, B. S.: Stresses around horizontal boreholes drilled in sedimentary rocks, *Journal of Petroleum Science and Engineering*, 2, 349–360, https://doi.org/10.1016/0920-4105(89)90009-0, 1989.

Amadei, B.: Rock Anisotropy and the Theory of Stress Measurements, Springer-Verlag, Berlin, New York, 1983.

Andrews, L. C., Catania, G. A., Hoffman, M. J., Gulley, J. D., Lüthi, M. P., Ryser, C., Hawley, R. L., and Neumann, T. A.: Direct observations of evolving subglacial drainage beneath the Greenland Ice Sheet, *Nature*, 514, 80–83, https://doi.org/10.1038/nature13796, 2014.

Catania, G. A. and Neumann, T. A.: Persistent englacial drainage features in the Greenland Ice Sheet, *Geophysical Research Letters*, 37, L02501, https://doi.org/10.1029/2009GL041108, 2010.

Covington, M. D., Gulley, J. D., Trunz, C., Mejia, J., and Gadd, W.: Moulin Volumes Regulate Subglacial Water Pressure on the Greenland Ice Sheet, *Geophysical Research Letters*, 47, e2020GL088901, https://doi.org/10.1029/2020GL088901, 2020.

Flowers, G. E.: Modelling water flow under glaciers and ice sheets, *Proceedings of the Royal Society A*, 471, 20140907–20140907, https://doi.org/10.1098/rspa.2014.0907, 2015.

Flowers, G. E. and Clarke, G. K. C.: A multicomponent coupled model of glacier hydrology 1. Theory and synthetic examples, *Journal of Geophysical Research*, 107, 2287, https://doi.org/10.1029/2001JB001122, 2002.

Goodman, R. E.: Introduction to Rock Mechanics, 2nd ed., Wiley, New York, 1989.

Gulley, J. D., Benn, D. I., Screaton, E., and Martin, J.: Mechanisms of englacial conduit formation and their implications for subglacial recharge, *Quaternary Science Reviews*, 28, 1984–1999, https://doi.org/10.1016/j.quascirev.2009.04.002, 2009.

Hoffman, M. J., Perego, M., Andrews, L. C., Price, S. F., Neumann, T. A., Johnson, J. V., Catania, G., and Lüthi, M. P.: Widespread Moulin Formation During Supraglacial Lake Drainages in Greenland, *Geophysical Research Letters*, https://doi.org/10.1002/2017GL075659, 2018.

Krawczynski, M. J., Behn, M. D., Das, S. B., and Joughin, I.: Constraints on the lake volume required for hydro-fracture through ice sheets, *Geophysical Research Letters*, 36, L10501, https://doi.org/10.1029/2008GL036765, 2009.

Lüthi, M. P., Funk, M., Iken, A., Gogineni, S., and Truffer, M.: Mechanisms of fast flow in Jakobshavn Isbrae, West Greenland: Part III. Measurements of ice deformation, temperature and cross-borehole conductivity in boreholes to the bedrock, *Journal of Glaciology*, 48, 369–385, https://doi.org/10.3189/172756502781831322, 2002.

Meierbachtol, T. W., Harper, J., and Humphrey, N.: Basal Drainage System Response to Increasing Surface Melt on the Greenland Ice Sheet, *Science*, 341, 777–779, https://doi.org/10.1126/science.1235905, 2013.

Paterson, W. S. B.: Secondary and tertiary creep of glacier ice as measured by borehole closure rates, *Reviews of Geophysics*, 15, 47–55, https://doi.org/10.1029/RG015i001p00047, 1977.

Poinar, K. and Andrews, L. C.: Challenges in predicting Greenland supraglacial lake drainages at the regional scale, *The Cryosphere*, 15, 1455–1483, https://doi.org/10.5194/tc-15-1455-2021, 2021.

Poinar, K., Joughin, I., Lenaerts, J. T. M., and Broeke, M. R. V. D.: Englacial latent-heat transfer has limited influence on seaward ice flux in western Greenland, *Journal of Glaciology*, 1–16, https://doi.org/10.1017/jog.2016.103, 2016.

Poinar, K., Joughin, I., Lilien, D., Brucker, L., Kehrl, L., and Nowicki, S.: Drainage of Southeast Greenland Firn Aquifer Water through Crevasses to the Bed, *Frontiers in Earth Science*, 5, https://doi.org/10.3389/feart.2017.00005, 2017.

Priest, S. D.: Discontinuity Analysis for Rock Engineering, 1st ed., Chapman & Hall, London; New York, 1993.

Schoof, C.: Ice-sheet acceleration driven by melt supply variability, *Nature*, 468, 803–806, https://doi.org/10.1038/nature09618, 2010.

Sergienko, O. V.: Glaciological twins: basally controlled subglacial and supraglacial lakes, *Journal of Glaciology*, 59, 3–8, https://doi.org/10.3189/2013JoG12J040, 2013.

Smith, L. C., Chu, V. W., Yang, K., Gleason, C. J., Pitcher, L. H., Rennermalm, A. K., Legleiter, C. J., Behar, A. E., Overstreet, B. T., Moustafa, S. E., Tedesco, M., Forster, R. R., LeWinter, A. L., Finnegan, D. C., Sheng, Y., and Balog, J.: Efficient meltwater drainage through supraglacial streams and rivers on the southwest Greenland ice sheet, *PNAS*, 112, 1001–1006, https://doi.org/10.1073/pnas.1413024112, 2015.

Stevens, L. A., Behn, M. D., McGuire, J. J., Das, S. B., Joughin, I., Herring, T., Shean, D. E., and King, M. A.: Greenland supraglacial lake drainages triggered by hydrologically induced basal slip, *Nature*, 522, 73–76, https://doi.org/10.1038/nature14480, 2015.

Turcotte, D. L. and Schubert, G.: Geodynamics, Cambridge University Press, 484 pp., 2002.

Vaughan, D. G.: Tidal flexure at ice shelf margins, *Journal of Geophysical Research*, 100, 6213–6224, https://doi.org/10.1029/94JB02467, 1995.

Weertman, J.: Theory of Water-Filled Crevasses in Glaciers Applied to Vertical Magma Transport beneath Oceanic Ridges, *Journal of Geophysical Research* 76(5), 1171–1183, 1971.

Weertman, J.: Can a water-filled crevasse reach the bottom surface of a glacier? *IASH Publications*, 95, 139–145, 1973.

Weertman, J.: Dislocation based fracture mechanics, *World Scientific*, London, 1996.

---

## Author Comment (AC2)

Author response to

**Interactive comment on "Controls on Greenland moulin geometry and evolution from the Moulin Shape model"**

Lauren C. Andrews, Kristin Poinar, and Celia Trunz
In review at *The Cryosphere*
September 1, 2021

**Response to Anonymous Referee #2**

Authors' responses are inline in blue text.

Apologies for taking a long time to produce this review.

This is a very interesting paper that presents a new model for the evolution of moulin geometry and explores how the results of this model for moulin shape and water level depend on various model parameters. It is argued that moulins comprise a sizeable fraction of the englacial-subglacial drainage system in Greenland, and that the time-evolution of their volume is a potentially important feature to include in englacial/subglacial models, offering improvements over a model that assumes a static moulin volume.

The study is an interesting one and I believe it deserves publishing in some form. However, I do have quite a lot of detailed questions, and some concerns, about the ingredients that go into the model. I will focus this review largely on these model details, from section 2 of the paper. Some of these may be sorted out by clarification as to what equations have actually been solved. As a general comment, there appears to be quite a lot of duplication of notation, which overcomplicates the presentation of the model and causes some confusion. I think it would also be helpful to express the physics in terms of differential equations rather than discrete increments that implicitly include time-steps.

We thank the reviewer for a thorough and helpful review, particularly the close reading of our mathematical formulations. Overall, we make an effort to revise the equations to ensure consistency and simplicity. We also reduce the number of subsections (there were a lot!).

In response to comments below and from Reviewer 1, we now provide several simplifying options in the model code and test their impact on model results in a new suite of sensitivity runs. This sensitivity study examines the impact of using (1) a fixed cylinder geometry, (2) a circular vs egg geometry, (3) elastic deformation calculated with or without surface deviatoric stresses, (4) a fixed subglacial channel size, (5) different subglacial channel lengths (i.e., differing hydraulic gradients), and (6) different forms of subglacial base flow. While not as exhaustive as our parameter sensitivity study, we believe that these additions provide insights into what components of the model are important to moulin water level and moulin shape, and what components can be used in a simplified form.

We respond to each comment below.  We also want to note that we have carefully considered the reviewer's comments on our application of elastic deformation. We still conclude that elastic deformation is an important component of moulin shape evolution, and we now make a concerted effort to better describe our parameterization of elastic deformation both throughout the main text and in a new Supplement. We believe that these improvements clarify the results and clarify the comparison between elastic deformation, viscous deformation, and phase change in the Discussion and in several figures.

**Major comments**

Section 2.2.1 - the rationale for modelling the moulin cross-section with this strange egg shape was weak for me.  It significantly complicates the model to do this, rather than to assume it has a circular cross-section, and it was not at all clear to me that there was any great advantage in doing so.  It is also not clear how r_1 and r_2 are separately evolved, and this needs to be made clearer. I ended up with the impression that the difference is likely because the open channel flow above the water line gives rise to a change in one of these but not the other; but below the water-line it seemed that r_1 and r_2 would evolve identically and therefore stay the same, assuming they start the same?  However, this should be made clear by telling us what exactly are the equations that govern the evolution of r_1 and r_2.  I would, at the same time, encourage the authors to think about simplifying things and assuming circular symmetry, since I think many of the results would still apply, and I think it would give a model that is more likely to be adopted by others.

> The reviewer makes a good point about our choice of horizontal coordinates. As deduced, the primary reason we chose an 'egg' geometry is to adequately represent the evolution of the moulin above the water line, however, with the goal of preserving the chosen model geometry while permitting simplicity, we have 1) included a description of why we chose a 'egg' geometry in the *Moulin shape coordinate system* section, 2) included a comparison of water level and moulin water storage in a new set of model runs (see our introductory response above), and 3) included the necessary module and run script changes in MouSh to allow users to select the desired geometry (see our Github repository and the *Model description* section).

Section 2.2.2 - I had great difficulty following the treatment of elastic deformation, and am slightly concerned that this is not dealt with correctly.  In particular, a number of figures (figure 6, figure 9) compare viscous and elastic 'deformation' as a *rate*, with units m/d.  Elastic deformation is not a rate - it is an instantaneous deformation and it results in a displacement (relative to some reference state) that is fixed, for fixed stress - in this context, that is the change in radius given by (4).  Presumably this must be viewed as relative to some 'reference' radius that can evolve in time due to viscous deformation and phase change.  The elastic displacement in (4) does itself evolve in time due to changes in water pressure, and therefore gives rise to a deformation rate that is d(\Delta r_E)/dP * dP/dt, i.e. proportional to the rate of change of water level, and perhaps that is what is being plotted in these figures, but I did not really have this impression. If that is indeed what is meant, note that the elastic deformation rate depends on the rate of change of P, not on P itself, so whether the pressure is above or below overburden is irrelevant to the sign of the deformation rate (it is instead a question of whether P is increasing or decreasing).

The reviewer is correct. We did not initially realize this subtlety (we're really calculating dR_E/dP * dP/dt) until pointed out here. Thank you. We now include a clarification of this point in the *Elastic deformation* section and within the derivation of elastic equations in the Supplement. (Note we also include a dR_E/da * da/dt term, where "a" is the moulin radius as calculated by the other modules.) The elastic Supplement includes an extended description of the assumptions made, the derivation of the elastic equations, including versions with zero deviatoric and shear stresses, and our parameterization of instantaneous elastic deformation over our model timestep (the information plotted in Figures 6 & 9).

This concern is tied up with the question above of how exactly r_1 and r_2 are evolved.  It seems to me that you would want to have 'reference' values of these that evolve according to the viscous processes; they satisfy an equation of the form dr/dt = melt-back - viscous closure (very similar to the subglacial channel in (28)); and then you want to add the elastic deformation given by (4) on top of those evolving reference values to get the actual radius at any instant in time.

Indeed, we do evolve the moulin radius by using a reference state to which the phase change, viscous deformation, and elastic deformation are added after each time step. In the revised manuscript, we made an effort to express the equations in the text to be uniform and to clearly describe the use of this reference state. To include elastic deformation, we re-express elastic deformation as a 'rate' according to Eqn S9.

In equation (4), I would be inclined to simply ignore the deviatoric stresses, which I expect are relatively small in most cases compared to the effective pressure P (it was not clear to me what you have actually assumed for them in the examples).  Given that you are comparing with a null model which contains no moulin physics whatsoever, I think there is some advantage in not making this one overly complicated!   Note that there are in any case some missing brackets in this equation.  In equation (5), the first term is presumably set to zero for z larger than h_w (i.e. above the water line)?  This equation could be made more consistent with (6), which is essentially the same thing, but where P is now called sigma_z.  (6b) should again by zero for z larger than h_w, I think.

We thank the reviewer for detailed consideration of the model equations. We reformulate equation 4 to provide the initial and time evolution of the elastic radius and include an option without deviatoric stresses: $\Delta r_E = \frac{r(1+\nu)}{Y}P$, in the Supplement. We agree that in our current model, deviatoric stresses can over-complicate the model and their values can be difficult to constrain. To examine the impact of deviatoric stresses, we complete a comparison between the model with and without deviatoric stresses in the elastic module as part of the new sensitivity study. We now include this option in the model code.

Equation (9) is a strange way of discretising the time-derivative and this is where confusion starts to arise as to how r is actually evolved, because this gives an incremental change in r (both r_1 and r_2 ?) due to only viscous processes, and it is not clear how this is combined with the changes due to phase change and elastic deformation.  The viscous closure of a moulin due to (7) is essentially identical to that for a subglacial channel as described in (27) and as described by Nye (1953) for the closure of a borehole.  I think it would be helpful to express it as a contribution to the time-derivative dr/dt, as (effectively) done in (27).

We rewrite Equation 9 as $\frac{dr}{dt} = \frac{r}{dt}(e^\epsilon - 1)$(we have also substituted $\epsilon = \dot{\epsilon} dt$), although this requires the RHS to have a time differential (dt) without a corresponding radial differential (dr). We note that this was the case in Catania and Neumann (2010), although their variable *t* is really a time interval (~dt), so it is hidden.

Section 2.2.2.2.2 (I don't think I've ever seen quite so many subsections!)

We eliminated two levels by removing the remote sensing section in response to Reviewer 1 and simplifying the ordering. The maximum subsection depth is now three and much more manageable.

The downstream deformation of the ice is interesting, but it wasn't clear to me how it is incorporated into the model. It seems like it translates the 'centreline' of the moulin? But doesn't affect r_1 and r_2? So does it actually have any effect on the rest of the model or is it just relevant for the visualisations like in figure 8? The formula in (10) assumes no slip at the bed, which is presumably not always going to be the case?

The shear deformation does not affect the moulin radius, so it is (as the reviewer says) just relevant for visualization. We discretize the moulin as a stack of planes with holes in them; one can see how shear between adjacent planes does not affect the radii of the holes. We add a sentence to clarify this point.

Basal slip is likely in the real world, and in a complete model of the englacial and subglacial system, basal slip could be relevant - though perhaps really only on timescales longer than the one melt season we model here. Our current parameterization does not include slip at the bed. Our coordinate system starts at the bed: the origin is where the uphill wall of the moulin meets the bed. We add text clarifying this in the *Shear deformation* section.

Section 2.2.3 - I was a bit confused why melting and refreezing are treated separately - you could simply write down an energy balance that allows for either to happen automatically, depending on the relative magnitude of turbulent heating and the conduction into the ice, without having to have any 'switch' between melt season and not. In (11), I would have thought that the dT/dx should really be a dT/dr, i.e. the radial temperature gradient away from the (roughly) cylindrical moulin; the distinction between them is quite important because conduction around a point source in two dimensions (ie. in the x,y plane) is very different from conduction in one dimension (i.e. in x alone). That said, solving the heat equation in the ice for each different z seems a lot of work for a model of a single moulin, and I wonder if a reasonable approach would be to simply *estimate* the temperature gradient at the moulin wall, dT/dr, as (\Delta T)/r_m, where \Delta T is the temperature difference to the far-field ice and r_m is the moulin radius. That would be consistent with the way you incorporate the estimate of sensible heat in (18) when considering melting.

We choose to separate freezing and melting primarily because we initially developed the model with only melting, following many subglacial models, but then included refreezing. While we see the point of the reviewer's comment, we choose to keep them separate, so that they retain consistency with the current, widely used subglacial equations. However, we do appreciate the reviewer's comment about the temperature gradient and make the relevant changes in our model code and manuscript.

The salient question - in regard to the temperature gradient across the monulin's radius - is how long does refrozen ice persist inside the moulin. In a calm environment with no summertime melting, refrozen ice would persist arbitrarily long, and the refreezing rate would drop over time as simultaneously (a) the refrozen layer thickened, and (b) the latent heat conducted into the ice sheet warmed the ice, lowering dT/dx (or dT/dr). On the other hand, in a turbulent environment with significant summertime melting, each winter's refrozen ice may quickly melt away, exposing cold ice to the moulin water. This should allow a fairly uniform pattern of refreezing each winter (Eqn. 8-9 in Alley et al., 2005). In our original paper, we incorrectly assumed the first scenario. In fact, the thickness of refrozen ice (~0.5 m over one winter) is much less than the thickness of ice melted off the moulin wall in summer (~0.1 m/day). Thus, the second scenario applies, and we now calculate the refreezing rate based on a simple equation (Eqn. 8-9 in Alley et al., 2005 and Eqn 13 in the *Refreezing* section of our revised manuscript) proportional to the square root of time elapsed since the end of the turbulent melt season (start of the refreezing season). Note that in response to Reviewer 1, we indicated that we would remove this parameterization; however, upon consideration of this comment, we now choose to retain it.

Section 2.2.4 - Equation (22) needs to include Q_base, similarly to equation (24). In fact, there seems to be some inconsistency and duplication between (22), (24) and (29). These equations are all expressing mass conservation, and (22) and (29) are really the same equation (I assume that the m in (29) must include the freezing rate -delta as well). But I think they should include Q_base if you're going to include Q_base in (24). And I think (24) should really include some terms to account for the rate of change of the cross-sectional area (it comes from inserting V_m as the integral of A_m from 0 to h in (29)).

Yes! Eqn 22 should include base flow and now it is consistent with Eqn 24; Eqn 29 now includes refreezing (this was an oversight in the original manuscript). We made an effort to revise, simplify and ensure consistency across all equations.

Section 2.2.4.2 - I think it would help to have a schematic picture of the moulin and the subglacial channel showing some of the various variables. It is slightly frustrating - but I can see that it may be unavoidable - to have the moulin shape model coupled so tightly to a subglacial channel model; ideally you'd like to be able to model the moulin separately. In this case, it seems that the subglacial channel is assumed to run from the bottom of the moulin to the ice-sheet margin, along which length the channel cross-section would presumably vary in reality, but I think that you assume a single value of S (the value at the bottom of the moulin?) is sufficient to describe how the flow evolves? This seems a reasonable simplification here, but I think could be explained a bit better, and as I say, a diagram might help. The 'b' in the hydraulic gradient on line 339 seems to disappear when this term is inserted in (28). The diagram might also help to explain Q_base, Q_in and Q_out. The use of Q_base seems fine to me, as for most moulins there will likely be water arriving at the bottom of the moulin from upstream as well as via the moulin.

All great points. We now include a detailed schematic of the model as a part of Figure 1. It includes a visual description of model terms and variables.

We initially developed MouSh with a fixed radius subglacial channel; however, identifying an appropriate constant subglacial channel radius was an impossible exercise, since the channel size significantly affects moulin water level/variability and could not be used for an entire melt season. For this reason, we needed to include an evolving subglacial channel. We share the reviewer's frustration about not being able to isolate the impact of moulin variability, so we now include a model run with a fixed subglacial channel geometry in the new sensitivity study.

Subglacial channel cross-sectional area is taken to be representative of the channel halfway between the moulin and terminus - this is now clarified in the model schematic. We acknowledge that the simplified subglacial channel model is not fully realistic, but feel it is a reasonable middle ground between a non-evolving subglacial channel (see problems listed above) and a full subglacial model (which would shift the focus and free parameters away from the moulin and into the subglacial system). Coupling with a full subglacial model is part of our future plans.

We are happy that the reviewer finds our $Q\_base$ parameterization adequate. We do include a small suite of different base flow types on moulin geometry in response to Reviewer 1.

Section 3 - The results section focuses a lot on parameter sensitivity, and it is great that this has been explored so thoroughly, but I found this hard to follow without it having first been outlined some of the general behaviour of the model. In particular, I think it would be helpful to see some sort of figure showing the periodic states to which the moulin apparently evolves. Just the fact that the modelled moulin approaches an 'equilibrium' does not seem an obvious result, and I think that equilibrium could be described a bit more fully. Presumably it involves the water level moving up and down on a diurnal timescale, and the moulin opening and closing? It would also be useful to know how this depends on the moulin input $Q\_in$ (for me that would seem more of interest than dependence on drag parameters etc, which we don't know very well). It seems quite surprising to me that if such an equilibrium is really reached, it depends on the initial moulin radius. Also, has the moulin model been run over the course of multiple years (with melt season and a winter), and how does it behave? This has implications for what an appropriate 'initial' moulin radius is, presumably. I think it would be helpful to have some general discussion along these lines, and figure(s) (perhaps like figure 6 or 8) that show the general behaviour of the model, before going into detail about how certain outputs depend on the parameters, since it would help give those more context.

Great point: the result that the moulin approaches a quasi-equilibrium state (with "quasi" to denote the diurnal fluctuations around a steady mean state) could be a significant result. This quasi-equilibrium state is not dependent on the initial moulin radius (figure below). We now make sure to explicitly state that the moulin's quasi-equilibrium state is independent of the initial given geometry.

[Figure]

We now include sections in the Results and Discussion that highlight the quasi-equilibrium state reached by the moulin, including how the characteristics of Q_in affect that state. As part of this, we include a figure highlighting how the magnitude and diurnal range of Q_in impact the moulin water level and geometry. We also include a new figure that examines the behavior of the model components. In summary, when we vary the absolute magnitude of Q_in but not its diurnal range (panel 1 below), we find that the quasi-equilibrium water level decreases, as does the diurnal variability (panel 2 below). This occurs in concert with an increase in the major moulin radius at the equilibrium water level (panel 3 below). The geometry and water level adjust such that elastic and viscous deformation are generally equal to melting over a given day.

[Figure]

We find that increasing the diurnal range of Q_in (panel 1 below) results in an increase in the quasi-equilibrium water level and its daily range (panel 2 below). Interestingly, the moulin radius at the mean water level is generally very similar, though with small diurnal differences, suggesting that the moulin exhibits a relatively constant geometry even with different diurnal ranges (panel 3 below). The only exception to this is when Q_in has no diurnal variability, due to the differences between water-filled and open-channel turbulent melting. These results indicate that Q_in does affect the quasi-equilibrium state of the moulin, but in a predictable fashion. Further analysis is included in the revised manuscript.

[Figure]

We stand by the importance of testing sensitivity to uncertain parameters like the drag coefficients. We found that modeled moulin geometry, unfortunately, has high sensitivity to the most uncertain parameters (drag coefficients, especially above the water line). This is an important result because it suggests the importance of moulin exploration above the water line in order to constrain moulin shape and water flow there.

We do not run the model over the winter. The quasi-equilibrium state is independent of the initial moulin radius and there are a number of complexities to explore with regard to winter freezing, creep closure, and re-opening (e.g. Catania & Neumann, 2010). We hope to explore this in a subsequent manuscript.

Figure 6 - see my earlier comments about comparing elastic and viscous deformation. I just don't understand what is actually plotted in panel f and g. Could you express whatever quantity is being plotted in terms of variables in the equations? Similarly for figure 9, and the associated discussion in section 4.5

> We now include a Supplement that describes how we take an instantaneous term and apply it over a timestep, making it comparable to viscous deformation in Figures 6/9. We also clarify this point in the *Elastic deformation* section. Both these changes allow for a better understanding of the comparison presented in the *The role of elastic deformation in moulin geometry* section.

**More minor comments**

L82 - why does taking k = 1 approximate likely channelized pathways? The usual thinking is that channels would tend to *lower* the water pressure and would therefore be associated with a lower value of k, if anything.

> This section has been removed to focus the manuscript more clearly on moulin geometry, in response to Reviewer #1.

Figure 1 is very nice. It might be noted that the elastic deformation here is quite different from all the other ones, in that the others are all *rates* - they accumulate every timestep to give continued deformation - whereas the elastic one is just static.

> We change the caption to briefly explain the difference between elastic deformation and the rest of the terms. We also include a new panel with a model schematic.

L185 - the small component of melting due to temperature differences between the water and ice seems to be ignored in the model, since it is later assumed that the water is at the melting temperature ?

> Yes, we ignore the sensible heat of the moulin water. This is justified because (a) it is small (sensible heat from 1°C is ~1/160 of latent heat), and (b) moulin water temperatures are not well known. We now state this explicitly in this section.

L255 - you seem to use both hydraulic diameter $D_h$ and hydraulic radius $R_h$ and it would keep the notation simpler to just work with one or the other.

> Done.

L262 - it sounds like in the end you take $f_R$ to be fixed (and vary it's value) so I wasn't sure what the point of introducing (16) was.

> We simplify this section by removing the equation and reference to the Bathurst parameterization of friction. We agree it is unnecessary because in all model runs, we keep the friction factor fixed at a user-defined value.

In (18) presumably S is really $A_m$, the moulin cross-sectional area?

> Yes. This was an error. We now replace S with $A_m$.

In (19) is $dh_l/dz$ the same as $dh_L/dL$ in (15), and is there significance in the change from lower case to upper case subscripts?

The reference to dh$_L$/dL on the line following the equation is in error, a holdover from a previous iteration. It should be dh$_L$/dz, where z is the vertical step height within the moulin and dh$_L$ is the path length from the center of each vertical step, taking into account the offset between moulin elements (see below for a sketch). We fix it and include a small diagram in Figure 1.

[Figure]

In (23), time appears to be in hours, not days.

Great point, hours it is. We have now corrected this.

In (24), h is the same as h_w ?

Yes it is; we have now corrected this and ensured that equations are consistent with Figure 1b.

Figure 7 - should there be a purple line in panel (d)?

The line for the 741 m moulin (purple) is hidden behind the 533 m moulin (black). We will clarify this in the caption.

L662 - I wasn't able to see this statement about the fixed moulin frequently overtopping the moulin in Fig 11a. How does the figure show this?

Hmm, we aren't sure about what we meant by referencing Figure 11 here. We revised the text and now reference a figure associated with the new set of sensitivity runs.

**References**

Alley, R. B., Dupont, T. K., Parizek, B. R., and Anandakrishnan, S.: Access of surface meltwater to beds of sub-freezing glaciers: preliminary insights, 40, 8–14, https://doi.org/10.3189/172756405781813483, 2005.

---

## Editor Decision (ED1)

Re. Manuscript `tc-2021-41`

Dear Dr Andrews and co-authors,

Thank you for your thorough response to the reviews and the substantially improved manuscript. The experiments are very clearly presented and the results easy to understand. I am now delighted to accept the manuscript for publication, subject to two minor amendments:

1. Can you change the acronym GIS for GrIS?
2. L272: The Gulley ref misspells the authors name (Gully vs. Gulley)

I would like to commend you for the easy availability of your code which really helped the review process.

Thank you for your patience with the review and editorial progress, I am so pleased that the manuscript can now be recommended for publication.

Best regards,

Liz

Dr Liz Bagshaw, Editor

---

## Author Response (AR2)

Author response to

**Review of "Controls on Greenland moulin geometry and evolution from the Moulin Shape model"**

Lauren C. Andrews, Kristin Poinar, and Celia Trunz
In review at *The Cryosphere*
April 15, 2022

**Response to Anonymous Referee #2**

Authors' responses are inline in blue text.

Thank you for addressing my previous comments on the manuscript. Many areas have now improved, including the focus on moulins rather than the subglacial system, the description of the elastic equations, and reporting of the sensitivity tests/results.

Thank you and thank you for the second careful review. We very much appreciate the attention to detail. Responding to your comments has improved the manuscript substantially. We, again, carefully considered how elastic deformation is treated in the model. This resulted in several changes to Supplement 1 and an update to elastic.m. We reperformed all experiments with updated elastic deformation, correct subglacial deformation, and a more realistic enhancement factor. The figures and text reflect these updated results. Although the overall conclusions are similar, there are differences: the substantial reduction in elastic deformation increases the time for the model to reach quasi-equilibrium; the correction to subglacial deformation results in an increase in diurnal variability within in the subglacial system and a decrease in diurnal variability within the moulin. These changes also minimized differences when compared to a static cylindrical moulin. The results and discussion have been revised to reflect these changes.

The figures, figure references, equations, and constants are now consistent. We apologize for any confusion arising from inconsistencies arising through the review process.

The line number issue is odd. We made the numbers smaller and tighter to the text to resolve this issue.

I have two major comments remaining, the first of which is again related to elastic deformation. The new explanation for elastic deformation is certainly clearer to follow and it makes much more sense to me now that you have added in equation 7 demonstrating how you convert from instantaneous elastic deformation to a rate based on the rate of change in pressure in the borehole. However, I was surprised that none of your figures or results changed following application of this method, and that you mentioned that this was how you were calculating elastic deformation in the initial round. I had a look at the code you provide in the git repository and from that it looks like you're still just plotting cumulative instantaneous elastic deformation rather than

taking the change in pressure into account (elastic.m, moulingeom_fcn.m and simpleplots.m). Here it seems like you calculate the absolute elastic deformation for each time step based on the current water level, and then use each of those timestep outputs as the deformation rate. Instead, according to your equations, you should be calculating the deformation rate based on the change in pressure since the previous time step, not the absolute pressure. From some quick calculations it looks like it should be about an order of magnitude smaller than you've plotted.

You are correct. We made a mistake. This comment forced us to reconsider the role of elastic deformation in moulin evolution (again) and revisit to our derivation. We made some modifications to Supplement 1 and the equations in the main text and reperformed all the model experiments with the correct formulation in elastic.m. Elastic deformation now has a negligible impact on moulin geometry and evolution (see Supplement S2.2.3). This result is discussed in Section 4.5.

Another potential issue I spotted in the online code is in moulingeom_fcn.m where c2 is listed as $1An^{-n}$ but in your manuscript it's $2An^{-n}$ (P11 line 18).

Fixed! Even though we feel a bit silly, this is the benefit of open code development. All model runs completed with this and elastic changes.

Section 4.5: In the first round of reviews I suggested that focusing on englacial void ratios while discussing the role of moulins in the hydrological system isn't directly useful and your response was that it puts your results in the context of adjacent ideas. My concerns still remain but I will cede that it is useful to include if you're clear that, if the time-varying and evolving moulins are used in a subglacial model, their capacity for storage could remove the necessity of applying an englacial void ratio. However, in lines 59-77 (P25) it seems that you're suggesting calculating an englacial void ratio for entire catchments based on parameterized moulin-like responses? I'm unclear how this would be achieved with the results that you've presented and this goes beyond what you are able to discuss in the manuscript. In general, as the subglacial model would only represent reality if the moulins were used as water inputs/storage features, having a catchment-wide englacial void ratio parameterization seems moot. This argument stands even if using a static moulin shape. Lines 59-76 therefore do not add to your argument so I suggest you remove them.

It was not our intention to suggest that we could parameterize the evolution of englacial void ratio. We removed the suggested lines and made some modifications to the text to soften the language and suggest that more work needs to be done before concrete conclusions can be drawn.

I list my line-by-line comments below but something happened with the manuscript line numbers which cutoff the first number. So from page 5 onwards I list the page number and associated line number that I can see.
Sorry. We fixed this issue in this round.

13: instead of 'and models melting', just 'and melting'
Done

19: repetition of 'representing'
Changed the first to 'Implementing'

98-99: you should refer to the Table where you list these values.
Done.

P5 53. You have F* as the enhancement factor but this is still listed as E in the table. You also need to define 'n'.
Changed E to F* and included n in the table. We also updated Table 1.

P6 81. Define Lf and ki
Done in text and in Table 1

P7 83. Using 'thus' doesn't follow here.
Removed.

P7 01. 'Although' instead of 'though'
Changed here and throughout the text.

P7 11 – Symbology for both ODE and PDE in this sentence.
Changed to ordinary on line 214.
P10 85. 'near the margins of the Greenland Ice Sheet'?
Added 'of the GIS margins'

P11 eq 27. You defined N on page 3 as Pw-Pi. Usually N is Pi-Pw and would be applied as such in equation 27 so you'll have to redefine it as something like Ns for the subglacial system. Alternatively change 'N' to 'P' for the moulin pressure as you have in the supplement.
We more carefully define P and N in the text and also add a line "Note that $P$ is not effective pressure, which is defined as $P = P_i - P_w$ (Cuffey and Paterson, 2010)" to clarify this point to readers.

P11 18. Lf should have been defined further up in the manuscript so should be removed here.
Checked all equations and defined constants when first used. In some instances, we define them a second time for clarity if it has been several equations since a constant was last used.

P12 40 This is confusing where you talk about steady diurnal inputs then switch to talking about seasonal and diurnal melt variability.
Completely right! Removed that sentence.

P12 53 Isn't this now deformation enhancement factor F*?
Yes, fixed.

P12 56. Which enhancement factor value did you choose when running these other tests and is there justification for it?

The Enhancement factors used are listed in the first sentence of the paragraph, but we add additional clarification about the range of enhancement factors expected in reality. "While the range of enhancement factors tested here cover a variety of ice conditions, including ice shelves and temperate glaciers, the Greenland ice sheet likely has values between 4 and 6 (e.g., Cuffey and Paterson, 2010). Outside of testing the model sensitivity to the enhancement factor, we assign $F^*$ a value of 5."

P13 81. The 2019 melt season? Above you say the 2015 melt season.

Apologies, not really sure where 2015 came from.

Section 2.5. Where are the corresponding figures for each of these sensitivity tests?

In each subsection, we now identify the location of the sensitivity results and associated figures

P14 05-08. Which parameters do you change for these tests and to what value? The choices are now more clearly defined in the main text and direct references to the subsections are listed.

P14 09-11. I don't understand how changing the supraglacial input to the moulin can test an evolving vs. fixed radius subglacial channel. The language here was unclear. It's now fixed and the description in the supplement is now directly referenced. Essentially, we used a simpler Qin when testing a fixed subglacial channel because it is extremely time consuming to find a fixed S that 'works' (e.g., the moulin isn't empty or always completely full).

P14 22. This should be 'mean moulin water level'

Fixed

P15 35. Do you mean a larger than expected moulin compared to other runs with smaller magnitude of input?

Yes, clarified.

P15 45. Not sure what Figure S2.4 g is.

Neither are we. Sorry! Fixed S5d

P15 48. Ice flow enhancement factor listed as E again. Also later in the paragraph.

Fixed.

P16 82. Is this E Youngs modulus or the ice flow enhancement factor?

Went through and did a through check and now everything should be correct.

P16 83-85. A lot of 'which's' in this sentence.

Completely fair, we eliminated most of them.

P17 97-00. It would be good to mention here again that the ice thickness and distance from terminus are proxies for hydraulic potential gradient which will be the main driver for these

parameters rather than assuming moulins in thicker ice always increases the moulin size and volume capacity. Done. Now we include this text: "These changes act to directly modify the hydraulic potential gradient of the moulin system. The modeled ice thicknesses and distances from the terminus result in hydraulic potential gradient variation between 0.009 and 0.2 (~180%)..."

P17 05. When you say you use standard values, do you mean the median values that are listed in Table 1?
Yes, changed to be clearer.

P17 06-07. This is repetition from the previous section.
It is, we modified this section to more clearly discuss the roles of different model components on moulin shape.

P18. 29. Although (also line 32). Check rest of manuscript for this.
We changed most instances of though to although.

P18. 38-43. Something strange going on with Figure numbers here. I think you mean Figure 6 when you say Figure 7 and Figure 8 when you say Figure 9?
Yes, I thought I checked this, but with the removal of the remote sensing section, the figure numbers shifted and the reference to the figures got fumbled. Figure references are now correct.

P18 41. 'Elastic' rather than 'ecstatic'.
Hahaha… fixed.

P18 53. I don't think you mean Fig 7m-n here.
Fixed.

P18. 61-62. 'To test this idea…' This is confusing coming at the end of this section. Move to the next section.
Deleted since there is an introductory sentence in the following section.

P19 74-77. Long sentence. Consider splitting.
Done

P19 88-91. Clarify here which moulin type dampens the supraglacial input signal, moulin water levels and increases subglacial channel size.
Now reads: "Changing moulin capacity…"

P20 97. A new paragraph would be helpful here.
Done, but this section was also modified due to changes in the elastic formulation.

P20 05-06. What inputs resulted in runaway growth or collapse?

This now reads:  Finally, we examine the impact of fixing the subglacial channel cross-sectional area $S$. Experimental results using a fixed $S$ and a seasonally evolving melt curve resulted unrealistically low or zero water levels during low, early season $Q_{in}$ and complete viscous collapse of the moulin if subglacial S was prescribed to be too large, or persistently high (always above the ice thickness) water levels and runaway moulin growth if subglacial S was prescribed to be too small. Therefore, we explore the impact of fixing $S$ using a constant mean $Q_{in}$ with an overlaid diurnal variability (Supplement Sect. S2.2.6).

P20 26. '…Maxwell time for ice, are more likely…'
Fixed.

P21 41. Remove one of the 'which's fixed
Fixed.

P21 44. Which differs from what?
Clarified.

P23 94. I'm not sure why 'nevertheless' follow from the previous sentences. Is that because Covington et al (2020) had a fixed geometry moulin? This could do with clarification.
We clarify this to read: (Covington et al. (2020) use a fixed moulin geometry), or that their measured water levels were not from the same moulin they mapped englacially. However, our results suggest that an evolving moulin capacity may be important to represent realistic moulin water levels. During much of the melt season, modeled water levels within an evolving moulin are lower and less variable than in a fixed, cylindrical moulin (Fig. 10).

P23 02. The primary closure mechanism.
Fixed, thanks.

P24 28. What do you mean 'the space of viscous and elastic parameter values'?
Clarified, but this section changed substantially.

Fig 1. Would be helpful to include the component symbol (e.g. ud, drv) in brackets after each colored label to make Figure 2 clearer.
Excellent point. Done.

Fig 3. Left hand side y axes should run 0 to 0.3 so the results are clearer. I mentioned this in my last review and your rebuttal was that it wouldn't leave room for the subpanel numbers. It would be better to have these subpanel numbers above each graph, or on the right-hand side to have the y-axis at a lower range.
Fair enough. We moved the panel lettering to the outside of the panels in Fig. 3 & 4. We also made an effort to improve the scaling on all axes and place the panel letters in a visible (though not always consistent) place.

Figure 3 and Figure 4. I'm unclear what the difference is between these figures. I see the differences in the subplots but from the captions it's not clear what they're each showing.
There was a mis-labeling in axes labeling and these are now corrected.

Fig 5 g. Would be helpful to have the moulin water depth also plotted in this figure to see how it relates to the phase change and deformation outputs. You also mentioned in your rebuttal that you now include the combined change in moulin shape in Fig 5 g, which I don't see here.
We updated Figure 5 to include the water levels (panel h). While we had initially proposed to include the total change but failed to include it, we now choose to not include it in order to prevent the plot from being too busy.

Fig 8. You say in the caption that all the grey lines are <1 but around day 35 it looks like they're above zero (as discussed in the text). I find it difficult to differentiate between the grey and purple lines in subpanel b. It would be clearer if the legend said 'Elastic:viscous ratio' and the y axis also refer to the ratio.
We removed the gray lines; with the modification of elastic deformation, it is substantially smaller than viscous deformation. We do keep the phase change: viscous ratio and change the y-label accordingly.

Fig 10. In the caption you say basin 1 is black and basin 2 is purple. They all look purple to me.
A remnant from the previous method of plotting. Now it is correct.

S1.2 why not have the hydrostatic stress in the same notation as the main manuscript (eq 3c)? This applies generally to the equations you present in the supplement.
We changed Eqs. 3a-c and the Supplement and main text now both use z.

S1.2.1 with v = 0.3, the difference for the plane stress solution is 9%.
This section was modified to test the impact of elastic deformation instead of surface stresses.

S2.2.2. Which moulin capacity is 47% smaller?
Fixed, the circular moulin is smaller (31%).

S.2.3 'Though' at the beginning of the sentence in the second paragraph is confusing – would be better if the sentence started with 'Overall'.
Fixed.

S2.2.4 'Reducing L reduces' should be rephrased.
Fixed.

---

## Author Response (AR3)

Re: Manuscript tc-2021-41

2 May 2022

Dear Dr. Bagshaw,

Thank you for the kind words.

We made the corrections indicated, added a missing word, and removed several instances of incorrect double spaces.

Best wishes,
Lauren Andrews
Kristin Poinar
Celia Trunz